# LiRA: Linguistic Robust Anchoring for Cross-lingual Large Language Models

## Abstract

As large language models (LLMs) rapidly advance, performance on high-resource languages (e.g., English, Chinese) is nearing saturation, yet remains substantially lower for low-resource languages (e.g., Urdu, Thai) due to limited training data, machine-translation noise, and unstable cross-lingual alignment. We introduce LiRA (Linguistic Robust Anchoring for Large Language Models), a training framework that robustly improves cross-lingual representations under low-resource conditions while jointly strengthening retrieval and reasoning. LiRA comprises two modules: (i) Arca (Anchored Representation Composition Architecture), which anchors low-resource languages to an English semantic space via anchor-based alignment and multi-agent collaborative encoding, preserving geometric stability in a shared embedding space; and (ii) LaSR (Language-coupled Semantic Reasoner), which adds a language-aware lightweight reasoning head with consistency regularization on top of Arca's multilingual representations, unifying the training objective to enhance cross-lingual understanding, retrieval, and reasoning robustness. We further construct and release a multilingual product retrieval dataset covering five Southeast Asian and two South Asian languages. Experiments across low-resource benchmarks (cross-lingual retrieval, semantic similarity, and reasoning) show consistent gains and robustness under few-shot and noise-amplified settings; ablations validate the contribution of both Arca and LaSR. Code will be released on GitHub and the dataset on Hugging Face.

## 1 Introduction

Large language models (LLMs) have achieved remarkable progress across a wide range of natural language understanding and reasoning tasks. However, their performance remains heavily skewed toward high-resource languages such as English and Chinese, while performance on LRLs continue to lag far behind. This disparity primarily arises from long-tailed pretraining distributions (Conneau et al., 2020) (Figure 1a), insufficient or noisy parallel data (Khayrallah & Koehn, 2018), and unstable cross-lingual alignment (Wu & Dredze, 2020). Consequently, the direct application of LLMs to retrieval and reasoning tasks in LRLs frequently results in diminished performance and inconsistent behavior, thereby compromising the potential for achieving global inclusiveness.

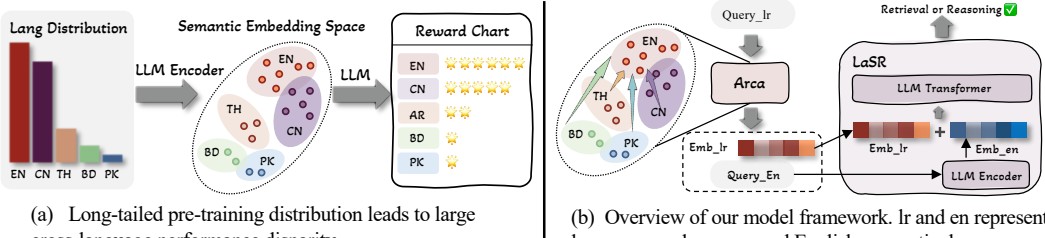

(a) Long-tailed pre-training distribution leads to large cross-language performance disparity.

(b) Overview of our model framework. lr and en represent low-resource language and English, respectively.

Figure 1: Challenge and overview of our LiRA.

Existing approaches to cross-lingual adaptation typically rely on machine translation (Artetxe et al., 2023; Shubham, 2024) or multilingual method (Singh et al., 2024; Mei & Zhao, 2025). While

translation-based pipelines have demonstrated a certain degree of effectiveness, they are inherently susceptible to error propagation and semantic drift, particularly in complex tasks involving multi-step reasoning or nuanced linguistic interpretation. On the other hand, multilingual encoders provide a degree of language-agnostic representation but lack the strong reasoning capability that LLMs demonstrate in English. Although prior work, such as MindMerger (Huang et al., 2024), has made progress in narrowing the performance gap of large language models in low-resource languages, its architectural design remains relatively simplistic and lacks grounding in a rigorous theoretical framework, limiting its generalizability and analytical depth. Similarly, Lusifer (Man et al., 2025) proposes a straightforward integration of a multilingual encoder with an LLM-based embedding model, primarily targeting embedding tasks rather than cross-lingual reasoning. Despite their empirical gains, both approaches remain theoretically underdeveloped. As a result, neither approach adequately closes the performance gap for low-resource languages across key tasks such as information retrieval, ranking, and reasoning.

Our work is motivated by the observation that current cross-lingual adaptation for retrieval still relies heavily on machine translation pipelines or multilingual encoders, which often suffer from translation noise and unstable cross-lingual representations, especially for low-resource languages in real-world e-commerce scenarios. These approaches typically lack an explicit treatment of how semantic drift and representation mapping errors propagate through the pipeline, making their robustness hard to reason about and to systematically improve. To address these challenges, we introduce **LiRA** (**Li**nguistic **R**epresentation **A**nchoring), a unified framework that anchors low-resource languages to the English semantic space while preserving LLM-level reasoning ability. Unlike prior approaches, LiRA is grounded in a rigorous theoretical foundation that certifies the completeness and stability of its cross-lingual representations (Figure 1b). LiRA integrates two complementary components: (i) *Arca*, which explicitly aligns multilingual representations with English through critic–actor interaction and feature anchoring, thus reducing semantic drift; and (ii) *LaSR*, a lightweight head that fuses multilingual and English embeddings with queue-based objectives, ensuring robust retrieval and reasoning under low-resource conditions. In this way, LiRA not only leverages the strong English capability of LLMs but also transfers it effectively to underrepresented languages.

Our main contributions are summarized as follows:

- We propose a novel cross-lingual reasoning framework that generalizes the strong English ability of LLMs to mid- and low-resource languages.

- We establish solid theoretical foundations, providing rigorous guarantees of LiRA's completeness and stability.

- We release a retrieval dataset covering 5 Southeast Asian and 2 South Asian mid- and low-resource languages, enabling further research in this underexplored area.

- Extensive experiments across ranking, retrieval, and reasoning tasks demonstrate that LiRA achieves new state-of-the-art performance.

## 2 RELATED WORK

**Cross-lingual Information Retrieval**  Early CLIR systems relied on multilingual PLMs (e.g., mBERT (Devlin et al., 2019), XLM-R (Conneau et al., 2020)) for alignment; recent work moves toward supervision-light transfer and LLM-embedding adaptation. LUSIFER integrates a multilingual encoder with an English-specialized LLM-based embedding model via a connector to yield strong zero-shot multilingual retrieval (Man et al., 2025). Beyond passage-level retrieval, CCPR formulates contextualized *phrase*-level cross-lingual retrieval to mitigate polysemy (Li et al., 2024), while XRAG benchmarks end-to-end cross-lingual RAG from retrieval to generation (Liu et al., 2025). Domain resources (e.g., CrossMath) and synthetic datasets (e.g., SWIM-IR) further broaden evaluation and training coverage (Gore et al., 2024; Thakur et al., 2023). Finally, cross-lingual contamination can inflate reported performance, calling for stricter data governance (Shi et al., 2024).

**LLM-based Cross-lingual Reasoning**  Recent studies (Cueva et al., 2024) have intensified attention to reasoning tasks in low-resource language settings. (Kim et al., 2024) MindMerger merges external multilingual understanding into LLMs and trains collaborative use of internal reasoning and external language skills, substantially boosting reasoning in non-English settings (Huang

et al., 2024). Process-level improvements—such as Chain-of-Preference Optimization and Chain-of-Code—provide transferable reasoning scaffolds that can be combined with language anchoring/forcing strategies (Liu et al., 2024; Zhang et al., 2024). Mechanism-focused studies reveal cross-lingual knowledge barriers and even "cross-lingual collapse" of reasoning traces toward dominant pretraining languages; mitigation includes mixed-language finetuning, reward shaping for language consistency, and representation steering to strengthen non-English token representations (Chua et al., 2024; Zhao et al., 2024; Park et al., 2025; Mahmoud et al., 2025). Recent surveys systematize multilingual reasoning evaluations and resources (Ghosh et al., 2025).

**LLMs, Machine Translation, and Evaluation.** Recent large language models(GPT-5 (Wang et al., 2025), Qwen (Yang et al., 2025)) achieve strong zero/few-shot multilingual transfer through pretraining and instruction tuning. Neural machine translation provides parallel or pseudo-parallel signals that help align cross-lingual representations, and translation quality metrics (automatic or learned) are commonly used to filter or weight supervision. (Team et al., 2022)

**Reinforcement Learning and Critics for Language Models.** Reinforcement learning (DeepSeek-AI et al., 2025) has been used to align language models via policy optimization and preference-based objectives. Actor–critic (Li et al., 2025) formulations supply learned critique/value signals and can stabilize training with replay/queue mechanisms. In multilingual representation learning, such critics can act as quality estimators over noisy supervision, reweighting updates to curb error propagation.

# 3 THEORETICAL FOUNDATIONS

## 3.1 PRELIMINARIES

**Setup.** Let $\mathcal{X}$ denote a LRL sentence space and $\mathcal{Y}$ the English sentence space. For any $y' \in \mathcal{N}_\delta(y)$ we set $\mathbf{z}^\star := [\, g(x);\, h(y') \,]$ (with the same $x$), so that the ratio compares $f_{\mathrm{LLM}}$ under $(y, y')$. $\mathcal{Y}_{\mathrm{data}}$ is the finite evaluation corpus on which neighborhoods and empirical quantiles are computed. Given a source sentence $x \in \mathcal{X}$, an LLM-based translator $T : \mathcal{X} \to \mathcal{Y}$ produces $y = T(x)$, and $y^\star \in \mathcal{Y}$ denotes a *perfect translation* that matches $x$ in semantic distribution (formalized below). We consider two representation paths: an *anchor map* $g : \mathcal{X} \to \mathbb{R}^d$ that embeds LRL sentences directly into the "English" semantic space, and an *English encoder* $h : \mathcal{Y} \to \mathbb{R}^d$. We form a concatenated cross-lingual representation $\mathbf{z} = [\, g(x);\, h(y) \,] \in \mathbb{R}^{2d}$, which is consumed by a downstream scorer $f_{\mathrm{LLM}} : \mathbb{R}^{2d} \to \mathbb{R}$ for retrieval or reasoning. Our goal is to certify the fidelity of $\mathbf{z}$ and provide measurable error bounds.

**Assumption 1** (Semantic anchoring). For all $x \in \mathcal{X}$, the mismatch between the anchor and the English encoding of its translation is bounded:

$$\|g(x) - h(T(x))\|_2 \le \epsilon_1, \qquad \epsilon_1 \ge 0.$$

**Assumption 2** (Translation fidelity). Let $s$ be a latent semantic variable with conditionals $p(s \mid x)$ and $p(s \mid y)$. The translator $T$ preserves semantics up to $\epsilon_2$ in KL:

$$D_{\mathrm{KL}}\big(p(s \mid x) \,\|\, p(s \mid T(x))\big) \le \epsilon_2, \qquad \epsilon_2 \ge 0.$$

**Definition 1** (RKHS representation). Let $h$ denote the English sentence encoder. We model $h(y)$ as the *kernel mean embedding* (KME) of the semantic distribution $p(s \mid y)$ into an RKHS $\mathcal{H}$ induced by a positive–definite kernel $k$:

$$h(y) \;=\; \mu_{p(s|y)} \;:=\; \mathbb{E}_{s \sim p(s|y)}\big[\varphi(s)\big], \qquad \varphi(s) \;:=\; k(s, \cdot).$$

The kernel embedding maps any probability measure $p$ over the semantic space into the RKHS specified by $k$ (i.e., $\mu_p = \mathbb{E}_{s \sim p}[\varphi(s)]$). For bounded inputs, the kernel satisfies

$$0 \;<\; k(s, s) \;=\; \big\langle k(s, \cdot),\, k(s, \cdot) \big\rangle_{\mathcal{H}} \;\le\; C^2,$$

for some constant $C > 0$. See A.1.2 for details.

**Definition 2** (Data-local Lipschitzness). On a finite discrete domain, any encoder admits a (local) Lipschitz constant. Concretely, over the dataset $\mathcal{Y}_{\mathrm{data}}$, we define the data-local Lipschitz constant

at $y$ (with neighborhood radius $\delta$) as

$$L_h^{\mathrm{loc}}(y; \delta) := \max_{y' \in \mathcal{N}_\delta(y)} \frac{\left\| f_{\mathrm{LLM}}(\mathbf{z}) - f_{\mathrm{LLM}}(\mathbf{z}^\star) \right\|_2}{\left\| \mathbf{z} - \mathbf{z}^\star \right\|_2}, \qquad \mathbf{z} = [\, g(x)\,;\, h(y)\,].$$

Here $\mathcal{N}_\delta(y) = \{\, y' \in \mathcal{Y}_{\mathrm{data}} : 0 < d_{\mathrm{tok}}(y, y') \le \delta \,\}$ denotes the token-edit neighborhood (we use $\delta = 1$ in most experiments). We also report the empirical $q$-quantile $L_h^{(q)}(\delta)$, where $q \in (0, 1)$ is the quantile level; $L_h^{(q)}(\delta)$ is the $q$-th empirical quantile of $L_h^{\mathrm{loc}}(y; \delta)$ over $y \in \mathcal{Y}_{\mathrm{data}}$. ; for example, with $q = 0.95$ we observe $L_h^{(0.95)} \approx 0.034$ (see A.1.2 for details.).

## 3.2 THEOREM

**Theorem** (Representation deviation)**.** Under Assumptions 1–2 and Definitions 1–2, let $\mathbf{z}^\star = [\, h(y^\star)\,;\, h(y^\star)\,]$ and $\mathbf{z} = [\, g(x)\,;\, h(y)\,]$. Then

$$\|\mathbf{z} - \mathbf{z}^\star\|_2 \;\le\; \epsilon_1 \;+\; C\sqrt{2\,\epsilon_2}\,. \tag{1}$$

**Corollary** (Downstream stability)**.** If $f_{\mathrm{LLM}}$ is locally Lipschitz with constant $L^{\mathrm{loc}}(y; \delta)$ as in Definition 2, then

$$\left\| f_{\mathrm{LLM}}(\mathbf{z}) - f_{\mathrm{LLM}}(\mathbf{z}^\star) \right\|_2 \;\le\; L^{\mathrm{loc}}(y; \delta)\left(\epsilon_1 + C\sqrt{2\,\epsilon_2}\right). \tag{2}$$

As $\epsilon_1, \epsilon_2 \to 0$, we obtain

$$\|\mathbf{z} - \mathbf{z}^\star\|_2 \to 0 \quad \text{and} \quad \|f_{\mathrm{LLM}}(\mathbf{z}) - f_{\mathrm{LLM}}(\mathbf{z}^\star)\|_2 \to 0.$$

In practice, we report empirical estimates, e.g., $L^{(0.95)} \approx 0.034$ and $C \approx 0.6866$ see A.1.2 for details. The theorem and its corollary guarantee that, under Assumptions 1–2, the model obtains high-fidelity (robust) representations that effectively support downstream tasks. Full math proof of theorem and corollary details are provided in A.1.

## 3.3 WHY CONCATENATE TWO REPRESENTATION PATHS?

Although feature concatenation introduces an additional error source compared to using a single feature vector, which appears to increase the overall error term in the model, we provide an an information-theoretic analysis showing that feature concatenation leads to more stable results. Model the two paths as noisy channels:

$$g(x) = s + \eta_g, \qquad h(y) = s + \eta_h,$$

with zero-mean, finite-covariance noises conditionally independent given $s$: $p(\eta_g, \eta_h \mid s) = p(\eta_g \mid s)\, p(\eta_h \mid s)$, where $\eta_g$ and $\eta_h$ are additive noises in the two channels, assumed zero-mean with finite covariance and conditionally independent given $s$. $\sigma_{s,k}^2 = \mathrm{Var}(s_k)$ is the variance of the $k$-th coordinate of the latent semantic vector $s$.

Define the information gain of concatenation:

$$\Delta I \;=\; I\big(s;\, [g(x), h(y)]\big) \;-\; I\big(s;\, g(x)\big).$$

where $I(\cdot\,;\,\cdot)$ and $H(\cdot \mid \cdot)$ denote Shannon mutual information and conditional entropy, respectively. By the chain rule and conditional independence,

$$I\big(s;\, [g, h]\big) \;=\; I(s; g) + I(s; h \mid g) \;=\; I(s; g) + H(s \mid g) - H(s \mid g, h) \;\ge\; I(s; g),$$

with equality iff $s \perp\!\!\!\perp h \mid g$. In cross-lingual settings, translation noise $\eta_h$ and anchoring noise $\eta_g$ are complementary, hence $I(s; h \mid g) > 0$ and $\Delta I > 0$. Therfore, if $\mathrm{Var}(\eta_{g,k}) \to \infty$ on some dimension $k$ (e.g., severe LRL ambiguity), then

$$I(s; g) \;\le\; \tfrac{1}{2}\log\!\Big(1 + \tfrac{\sigma_{s,k}^2}{\mathrm{Var}(\eta_{g,k})}\Big) \to 0,$$

The quantity in parentheses is the per-coordinate signal-to-noise ratio (SNR), defined as $\mathrm{SNR}_k = \sigma_{s,k}^2 / \mathrm{Var}(\eta_{g,k})$. For the additive channel $G_k = s_k + \eta_{g,k}$, the classical Gaussian-channel bound implies $I(s_k; G_k) \le \tfrac{1}{2}\log\!\big(1 + \mathrm{SNR}_k\big)$. while a stable English path ($\mathrm{Var}(\eta_{h,k}) < \infty$) yields a strictly positive lower bound for $\Delta I$ on that dimension. Hence the concatenation $\mathbf{z} = [g(x); h(y)]$ overcomes single-path bottlenecks, and the information gain offsets the apparent worst-case bound increase.

# 4 METHOD

## 4.1 ARCA

Let $\mathcal{X}$ be a low–resource source space and $\mathcal{Y}$ the English space. For any $x \in \mathcal{X}$, a translator $T : \mathcal{X} \to \mathcal{Y}$ yields $y = T(x)$. We introduce two representation paths: an *anchoring map* $g : \mathcal{X} \to \mathbb{R}^d$ that lands source sentences directly in an "English semantic" space, and an English encoder $h : \mathcal{Y} \to \mathbb{R}^d$. We concatenate $\mathbf{z} = [g(x); h(y)]$ and score it with an LLM head $f_{\text{LLM}}$. ARCA aims to reduce the two terms appearing in our generalization bound (Sec. 3.2): the *anchoring error* $\epsilon_1$ and the *translation distortion* $\epsilon_2$.

ARCA (Figure 2) comprises three modules: (i) a *Translation Critic* that judges candidates with semantic/emotional/pragmatic scores and (projected) embedding similarity; (ii) an *Embedding Critic* that anchors feature paths to translation paths via a regression-style penalty; and (iii) an *Actor* trained with policy gradients that fuses both critics.

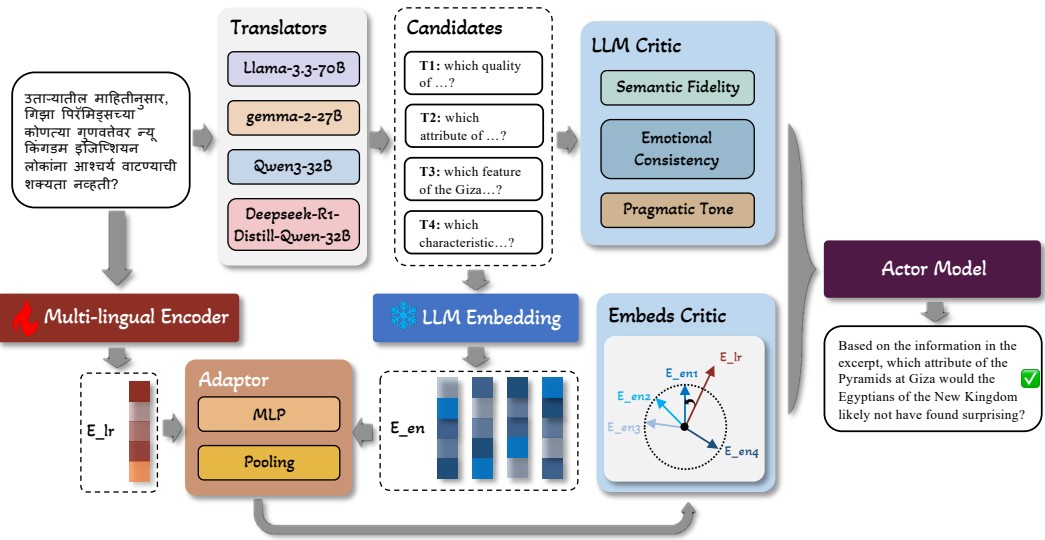

Figure 2: An overview of Arca.

### 4.1.1 EMBEDDING CRITIC AND FEATURE ANCHORING (MINIMIZING $\epsilon_1$)

On the query side, the input is constructed with a feature prefix of length $S_{\text{feat}}$ followed by text tokens of length $T$: $\mathbf{H}^{(q)} \in \mathbb{R}^{(S_{\text{feat}}+T) \times D}$. On the document (candidate) side we have $\mathbf{H}^{(d)} \in \mathbb{R}^{T \times D}$. We define the *feature-path* representation (average over the feature prefix)

$$\mathbf{g}_i = \frac{1}{S_{\text{feat}}} \sum_{s=1}^{S_{\text{feat}}} \mathbf{H}^{(q)}_{s,:}, \tag{3}$$

and the *translation-path* representation as pooled document states

$$\mathbf{h}_i = \text{pool}\big(\mathbf{H}^{(d)}\big), \tag{4}$$

where $\text{pool}(\cdot)$ (e.g., last-token/mean/attention) eliminates the tokenizer-induced length mismatch; the pooling dimension is a hyperparameter.

The *feature-anchoring loss* penalizes the discrepancy between the two paths:

$$\mathcal{L}_{\text{anchor}} = \frac{1}{B} \sum_{i=1}^{B} \left\| \mathbf{g}_i - \mathbf{h}_i \right\|_2^2. \tag{5}$$

*Role for $\epsilon_1$.* Eq. equation 5 directly contracts the anchoring radius $\epsilon_1$ by regressing the feature-path vector towards the translation-path vector.

We also define the similarity used in Eq. equation 10. Let $E_{\text{en},k} = \text{Emb}_{\text{LLM}}(y_k)$ be the English embedding sequence. After Adaptor projection (pool→MLP), we obtain a source anchor $\tilde{e}$ and a candidate vector $e_k$ in a shared space and compute

$$\text{sim}_k = \cos(\tilde{e}, e_k).$$

### 4.1.2 TRANSLATION CRITIC (MINIMIZING $\epsilon_2$)

Given a source $x$ and its candidate set $\{y_k\}_{k=1}^K$, a lightweight LLM judge produces three calibrated scores $s_k, e_k, p_k \in [1, 10]$ for *semantic fidelity*, *emotional consistency*, and *pragmatic tone*. We collect

$$\mathbf{r}_k = [\, s_k, \; e_k, \; p_k \,]^\top.$$

*Role for $\epsilon_2$.* These scores probe adequacy and well-formedness of $y_k$ with respect to $x$, serving as a proxy for small semantic divergence between $p(s\,|\,x)$ and $p(s\,|\,y_k)$; maximizing their contribution in the policy (see Sec. 4.1.3) drives smaller $\epsilon_2$.

### 4.1.3 ACTOR–ADAPTOR

**Adaptor.** The multilingual encoder and the LLM use different tokenizers ($L_x \neq L_k$) and embedding spaces. We first apply temporal pooling $P_r(\cdot)$ (window/stride $r$, a hyperparameter) and then an MLP to align both sides to a shared dimension $d_s$:

$$\tilde{\mathbf{u}} = \phi\big(W_{\text{mul}} P_r(\text{Enc}_{\text{multi}}(x)) + \mathbf{b}_{\text{mul}}\big), \quad \mathbf{v}_k = \psi\big(W_{\text{en}} P_r(\text{Emb}_{\text{LLM}}(y_k)) + \mathbf{b}_{\text{en}}\big). \tag{6}$$

We define the cross-space similarity

$$\text{sim}_k = \cos\big(\tilde{\mathbf{u}}, \mathbf{v}_k\big), \tag{7}$$

and add an encoder-alignment term on the chosen candidate $a$:

$$\mathcal{L}_{\text{enc}} = -\cos\big(\tilde{\mathbf{u}}, \mathbf{v}_a\big). \tag{8}$$

**Actor.** For each candidate we form the policy feature by concatenating the critic scores with the adaptor similarity:

$$\mathbf{c}_k = [\, s_k, \; e_k, \; p_k, \; \text{sim}_k \,]^\top. \tag{9}$$

A small MLP produces logits $g_\phi(\mathbf{c}_k)$ and the policy $\pi_\phi(k \mid \mathbf{c}_{1:K}) = \text{softmax}([g_\phi(\mathbf{c}_1), \ldots, g_\phi(\mathbf{c}_K)])_k$. We use the composite reward

$$R_k = 0.1 \cdot (\alpha s_k + \beta e_k + \gamma p_k) + \delta \, \text{sim}_k, \tag{10}$$

sample $a \sim \pi_\phi$, and optimize with REINFORCE:

$$\mathcal{L}_{\text{RL}} = -\mathbb{E}_{a \sim \pi_\phi}[R_a] \approx -\log \pi_\phi(a \mid \mathbf{c}_{1:K}) \cdot R_a. \tag{11}$$

The Actor–Adaptor objective used in training is $\mathcal{L}_{\text{AA}} = \mathcal{L}_{\text{RL}} + \eta \, \mathcal{L}_{\text{enc}}$, which complements the feature-anchoring loss in Sec. 4.1.1

### 4.1.4 OVERALL OBJECTIVE

The full ARCA objective is

$$\mathcal{L} = \mathcal{L}_{\text{RL}} + \eta \, \mathcal{L}_{\text{enc}} + \lambda \, \mathcal{L}_{\text{anchor}}. \tag{12}$$

Here, $\mathcal{L}_{\text{anchor}}$ reduces the anchoring error $\epsilon_1$, while $\mathcal{L}_{\text{trans}}$ favors low-distortion candidates, shrinking $\epsilon_2$—jointly tightening the bound in Sec. 3.2.

## 4.2 LASR

Given any-language text, a multilingual encoder yields $E_{\text{lr}}$ while a shared English encoder (prompted LLM encoder) yields $E_{\text{en}}$. $E_{\text{lr}}$ denotes the text embedding in the low-resource language, and $E_{\text{en}}$ denotes the embedding of the corresponding English text. We fuse $E_{\text{en}}$ and $E_{\text{lr}}$ into a single $\ell_2$-normalized embedding used for both ranking and retrieval. Training is supported by two FIFO buffers: (i) *CorrQueue* for correlation-based objectives under small batches, and (ii) *DocQueue* for listwise nDCG with in-language negatives. Both queues are stop-grad for cached entries and are updated in FIFO manner with a maximum size $K$.

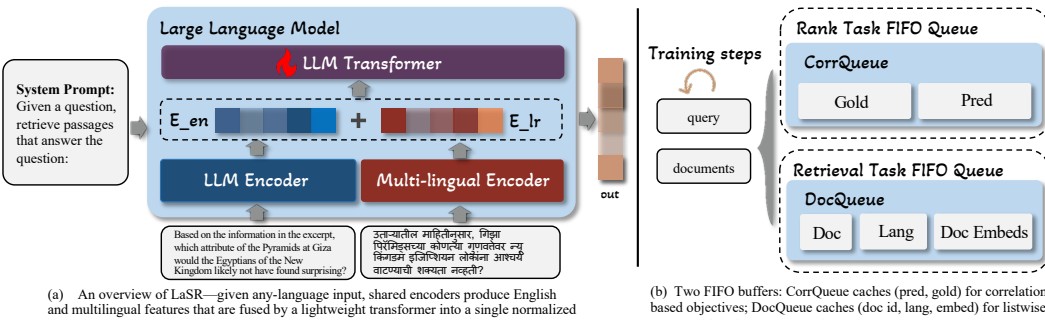

(a) An overview of LaSR—given any-language input, shared encoders produce English and multilingual features that are fused by a lightweight transformer into a single normalized embedding (q/d) for retrieval and ranking.

(b) Two FIFO buffers: CorrQueue caches (pred, gold) for correlation-based objectives; DocQueue caches (doc id, lang, embed) for listwise nDCG with in-language negatives.)

Figure 3: An overview of LaSR.

**Encoders.** Each branch follows "tokenize $\rightarrow$ encoder $\rightarrow$ pooling": $E_{\text{en}} = \text{pool}(\text{Enc}_{\text{LLM}}(\text{prompt}(x)))$ and $E_{\text{lr}} = \text{pool}(\text{Enc}_{\text{multi}}(x))$. Pooling is fixed (last-token or mean) and the two streams are concatenated and linearly projected before entering the LLM Transformer. The transformer attends across the two streams and returns a fused hidden vector $\mathbf{z}$, which is then normalized $\hat{\mathbf{z}} = \mathbf{z}/\|\mathbf{z}\|$.

**Training steps (Figure 3b).** Each optimization step processes two mini-batches: (1) query batch and (2) document batch. We forward them through the shared encoders and the LLM Transformer to obtain $\{\hat{\mathbf{z}}_q\}$ and $\{\hat{\mathbf{z}}_d\}$ and compute $s(q,d)$ for the task at hand. Two FIFO buffers are updated in the background:

- **CorrQueue (Rank task).** Caches tuples (Pred, Gold) produced on ranking datasets (e.g., STS). At step $t$ we concatenate the current predictions/labels with up to $K$ cached pairs (detached, no gradient) and compute a *correlation loss*: $\mathcal{L}_{\text{corrQ}} = \alpha\,(1 - \text{Pearson}) + (1 - \alpha)\,(1 - \text{Soft-Spearman})$. See A.4 for details.

- **DocQueue (Retrieval task).** Caches (doc_id, language, $\hat{\mathbf{z}}_d$) from recent steps for *in-language* hard-negative mining. Given a query, we form a candidate list (1 positive + mined negatives), compute differentiable ranks and a *soft nDCG@k* objective $\mathcal{L}_{\text{ndcg}} = 1 - \text{nDCG@}k$. To avoid mislabeled near negatives, we apply a temperatured down-weighting on very similar non-positives ("safe negatives"), and add two light regularizers (top-1 hinge, mean/variance control) to stabilize training: $\mathcal{L}_{\text{retr}} = \mathcal{L}_{\text{ndcg}} + \lambda_h \mathcal{L}_{\text{hinge}} + \lambda_r \mathcal{L}_{\text{mv}}$. See A.4 for details.

## 5 EXPERIMENT

### 5.1 EXPERIMENTAL DETAILS

All experiments are conducted on a single server with $4\times$ A100-80GB GPUs. We evaluate LiRA on three families of tasks to assess generality: (i) retrieval, measured by nDCG@10; (ii) sentence ranking, measured by Pearson correlation; and (iii) reading comprehension and mathematical reasoning, both measured by accuracy. For retrieval and sentence ranking we use Qwen3-Embedding-8B as the backbone encoder. For reading comprehension and mathematical reasoning we use Qwen3-4B as the backbone model. Model choices, full hyperparameters and additional implementation details are reported in the A.3. Each experiment was repeated 10 times, and we report the average as the final result.

Due to potential discrepancies in pretraining data, we ensured fairness by fine-tuning all models used in our experiments (including Qwen3) on each dataset with identical training hyperparameters before evaluation. Interestingly, we observed that fine-tuning Qwen3 brought no significant gains on existing public datasets, except for our newly introduced LazRetrieval dataset. This phenomenon may be attributed to Qwen3 having already seen portions of these public datasets during its pretraining phase.

Table 1: Evaluation on main public retrieval datasets. **bold** indicates the best result; underlined indicates the second best.

| Method | | MLQARetrieval | BelebeleRetrieval | STS22 | Avg. |
|---|---|---|---|---|---|
| SimCSE | *2021* | 7.41 | 18.35 | 37.95 | 21.24 |
| ST5-XXL | *2021* | 20.82 | 41.68 | 59.02 | 40.51 |
| GTR-XXL | *2021* | 20.19 | 38.02 | 60.11 | 39.44 |
| Contriever | *2022* | 9.75 | 22.94 | 41.72 | 24.80 |
| GTE-large | *2023* | 16.99 | 31.82 | 53.79 | 34.20 |
| BGE-en-1.5 | *2023* | 16.64 | 31.19 | 50.77 | 32.87 |
| E5-large | *2024* | 17.04 | 31.12 | 54.31 | 34.16 |
| E5-Mistral | *2024* | 31.54 | 54.75 | 71.37 | 52.55 |
| LUSIFER | *2025* | 36.68 | 57.81 | 70.49 | 54.99 |
| Qwen3-E-8B | *2025* | 81.13 | 85.94 | 71.64 | 79.57 |
| **LiRA** | **ours** | **81.66** | **87.03** | **75.00** | **81.23** |

Table 2: Evaluation on our new LazRetrieval dataset. **bold** indicates the best result; underlined indicates the second best.

| Method | | Bd | Id | My | Pk | Th | Ph | Vn | Avg. |
|---|---|---|---|---|---|---|---|---|---|
| Sentence-T5-XXL | *2021* | 34.11 | 71.77 | 49.19 | 27.84 | 23.58 | 84.20 | 28.61 | 44.56 |
| GTR-XXL | *2021* | 34.85 | 75.92 | 49.36 | 30.39 | 22.94 | 84.94 | 46.15 | 48.17 |
| SimCSE | *2021* | 31.76 | 66.71 | 43.27 | 27.68 | 32.32 | 74.00 | 44.16 | 44.88 |
| Contriever | *2022* | 39.95 | 74.95 | 48.90 | 35.71 | 15.43 | 83.74 | 64.75 | 51.00 |
| GTE-large | *2023* | 39.36 | 77.59 | 51.88 | 36.76 | 17.21 | 87.65 | 65.34 | 52.61 |
| BGE-en-v1.5 | *2023* | 41.06 | 78.78 | 53.28 | 37.52 | 18.35 | 88.18 | 68.72 | 54.09 |
| E5-large-v2 | *2024* | 41.04 | 78.78 | 53.77 | 37.22 | 17.63 | 87.24 | 61.31 | 52.84 |
| E5-Mistral-7B | *2024* | 48.27 | 75.43 | 71.01 | 53.62 | 61.75 | 83.18 | 65.44 | 64.51 |
| Qwen3-E-8B | *2025* | 65.97 | **79.18** | 80.56 | 65.39 | 82.48 | 86.59 | 77.31 | 76.78 |
| **LiRA** | **ours** | **66.30** | 78.53 | **81.54** | **68.53** | **83.12** | **87.44** | **78.48** | **77.71** |

## 5.2 DATASETS

We use standard public datasets: *BelebeleRetrieval* (Bandarkar et al., 2024), *MLQARetrieval* (Enevoldsen et al., 2025), *STS22* (Enevoldsen et al., 2025), *MGSM* (Shi et al., 2022), and *X-CSQA* (Lin et al., 2021). The first two are retrieval benchmarks, *STS22* evaluates sentence-level correlation, and *MGSM/X-CSQA* assess mathematical reasoning and reading comprehension, respectively. We additionally release a de-identified e-commerce retrieval dataset, LazRetrieval, and a larger companion set, LazRetrieval-mega. Both cover seven Southeast-Asian languages: Vietnamese (Vi), Thai (Th), Indonesian (Id), Malay (Ms), Urdu (Ur), Bengali (Bn), and Filipino/Tagalog (Ph). LazRetrieval contains **10 k** examples per language; LazRetrieval-mega contains **1,000 k** examples per language and is intended for pretraining/supporting large-scale adaptation. Unless otherwise noted, our experiments use *LazRetrieval*. Since *MGSM* and *X-CSQA* have no training split in our setup, we evaluate in a zero-shot fashion: the Arca is trained on *BelebeleRetrieval*, while the lightweight LaSR head is left untrained for these tasks. Detailed information about the datasets can be found in the A.2.

## 5.3 RESULTS ANALYSIS

**Retrieval & sentence ranking.** On public benchmarks (Table 1), LIRA consistently improves the base model (Qwen3-E-8B) on all three metrics: MLQARetrieval 81.66 vs. 81.13 (+0.53), BelebeleRetrieval 87.03 vs. 85.94 (+1.09), and STS22 75.00 vs. 71.64 (+3.36), yielding a higher macro average 81.23 (+1.66). On our new LazRetrieval-70K (Table 2), LIRA also improves the average from 76.78 to 77.71 (+0.93). The gains are particularly pronounced on relatively low-resource locales (e.g., pk +3.14, vn +1.17, my +0.98), suggesting that anchoring $g(x)$ to the English space

Table 3: MGSM accuracy (%). **Bold** indicates the best; underlined indicates the second best.

| Method | | Bn | Th | Sw | Ja | Zh | De | Fr | Ru | Es | En | Avg. |
|---|---|---|---|---|---|---|---|---|---|---|---|---|
| MonoReason | *2024* | 6.8 | 7.2 | 6.8 | 36.4 | 38.4 | 55.2 | 54.4 | 52.0 | 57.2 | 68.8 | 38.3 |
| MultiReason-Lora | *2022* | 29.6 | 35.2 | 28.0 | 52.0 | 54.8 | 59.6 | 58.4 | 62.4 | 59.6 | 64.8 | 50.4 |
| MultiReason-SFT | *2024* | 33.2 | 40.0 | 42.0 | 42.0 | 42.0 | 45.2 | 44.8 | 45.2 | 48.0 | 52.0 | 43.4 |
| QAlign | *2024* | 39.6 | 40.4 | 44.0 | 44.0 | 48.4 | 54.8 | 56.8 | 52.4 | 59.6 | 68.0 | 49.6 |
| LangBridge | *2024* | 42.8 | 50.4 | 43.2 | 40.0 | 45.2 | 56.4 | 50.8 | 52.4 | 58.0 | 63.2 | 50.2 |
| Translate-En | *2023* | 48.4 | 37.6 | 37.6 | 49.2 | 46.8 | 60.4 | 56.4 | 47.6 | 59.6 | 65.5 | 50.6 |
| MindMerger-Hard | *2024* | 46.0 | 36.0 | 48.4 | 52.4 | 54.4 | 60.4 | 56.0 | 60.4 | 62.0 | 71.2 | 54.7 |
| MindMerger-Soft | *2024* | 50.4 | 52.8 | 57.2 | 54.4 | 53.6 | 61.2 | 57.6 | 60.8 | 58.4 | 66.8 | 57.3 |
| Qwen3-4B | *2025* | 87.2 | 90.0 | **78.0** | 86.4 | 88.8 | **90.0** | 88.4 | **92.8** | **94.0** | **95.2** | 89.1 |
| **LiRA** | **Ours** | **88.4** | **91.2** | **78.0** | **86.8** | **89.6** | 89.6 | **88.4** | **92.8** | 92.4 | **95.2** | **89.2** |

Table 4: X-CSQA accuracy (%). **Bold** indicates the best; underlined indicates the second best.

| Method | | Sw | Ur | Hi | Ar | Vi | Ja | Pl | Zh | Nl | Ru | It | De | Pt | Fr | Es | En | Avg. |
|---|---|---|---|---|---|---|---|---|---|---|---|---|---|---|---|---|---|---|
| Translate-En | *2023* | 36.5 | 41.3 | 48.4 | 44.6 | 51.8 | 47.1 | 53.3 | 51.5 | 55.0 | 56.3 | 57.3 | 54.7 | 57.2 | 55.5 | 71.3 | 71.3 | 52.3 |
| MultiReason-Lora | *2022* | 25.1 | 32.0 | 39.2 | 42.2 | 56.6 | 55.9 | 60.6 | 62.2 | 61.3 | 62.8 | 66.3 | 64.9 | 66.2 | 67.4 | 67.7 | 79.3 | 56.9 |
| MultiReason-SFT | *2024* | 27.6 | 29.2 | 32.0 | 28.7 | 38.8 | 38.7 | 45.5 | 43.8 | 45.9 | 46.5 | 50.2 | 49.1 | 51.2 | 52.1 | 54.3 | 67.2 | 43.8 |
| MonoReason | *2024* | 24.2 | 25.1 | 32.9 | 32.3 | 50.9 | 49.1 | 50.6 | 56.5 | 57.5 | 56.0 | 56.0 | 61.2 | 61.7 | 63.5 | 64.0 | 76.3 | 51.3 |
| QAlign | *2024* | 35.1 | 32.6 | 37.8 | 36.3 | 50.5 | 49.2 | 57.1 | 54.8 | 56.3 | 58.3 | 58.3 | 58.8 | 59.8 | 60.3 | 63.1 | 75.7 | 52.3 |
| LangBridge | *2024* | 31.8 | 30.5 | 30.6 | 30.6 | 33.3 | 33.9 | 39.8 | 39.8 | 38.4 | 39.1 | 37.4 | 36.4 | 33.8 | 38.2 | 38.8 | 44.4 | 36.1 |
| MindMerger-Hard | *2024* | 33.1 | 29.9 | 40.4 | 37.7 | 52.9 | 49.9 | 54.7 | 55.4 | 58.0 | 59.7 | 58.6 | 61.9 | 62.5 | 63.6 | 75.2 | 75.2 | 53.1 |
| MindMerger-Soft | *2024* | **45.5** | 46.2 | 48.4 | 51.4 | 60.6 | 53.9 | **63.3** | 62.9 | 63.8 | **66.8** | **67.0** | **67.1** | **68.1** | **69.1** | 75.2 | 78.1 | 61.0 |
| Qwen3-4B | *2025* | 34.8 | 50.4 | 51.6 | 60.6 | 60.9 | 58.3 | 61.2 | 65.0 | 63.6 | 62.9 | 66.5 | 65.8 | 67.5 | 65.7 | 68.2 | 81.7 | 61.5 |
| **LiRA** | **Ours** | 35.3 | **51.1** | **52.1** | **61.2** | **61.6** | 58.7 | 61.9 | **65.7** | **64.0** | 63.0 | 66.1 | 66.2 | 67.6 | 66.2 | 68.9 | **82.5** | **62.0** |

and the LaSR head together reduce translation/representation noise. We also observe better rank-correlation (STS22), matching our design of queue-augmented CorrQ and listwise soft-nDCG.

**Mathematic.** On MGSM(Table 3), LiRA brings a small but consistent gain over Qwen3-4B: macro avg 89.2 vs. 89.1 (+0.1). Per-language analysis shows improvements or ties on $9/11$ languages (e.g., Bn/Th/Zh), while a few high-resource languages remain similar (De, Es). This indicates that the concatenated representation $[g(x); h(y)]$ improves reasoning robustness without hurting English performance.

**Comprehension.** On X-CSQA(Table 4), LiRA outperforms Qwen3-4B on 15/16 languages and raises the macro average from 61.5 to 62.0 (+0.5). Improvements concentrate on lower-resource or typologically distant languages (Ur/Hi/Ar/Vi/Zh), consistent with our motivation that concatenating $g(x)$ and $h(y)$ adds complementary information and mitigates information bottlenecks.

Table 5: Performance of different embedding models with and without LiRA.

| Method | MLQARetreival | BelebeleRetrieval | STS22 | Avg |
|---|---|---|---|---|
| GTE-large | 16.99 | 31.82 | 53.79 | 34.20 |
| GTE-large+LiRA | 21.43 | 38.29 | 59.17 | 39.63 |
| BGE-en-1.5 | 16.64 | 31.19 | 50.77 | 32.87 |
| BGE-en-1.5+LiRA | 21.01 | 35.64 | 53.94 | 36.83 |
| E5-Mistral | 31.54 | 54.75 | 71.37 | 52.55 |
| E5-Mistral+LiRA | 34.23 | 57.24 | 76.55 | 56.01 |

**Cross-backbone robustness.** Table 5 evaluates LiRA as a pluggable module on three representative encoders. Across MLQA Retrieval, BelebeleRetrieval, and STS22, LiRA consistently improves over the corresponding backbones. The averaged gains over the three tasks are positive for all backbones tested, suggesting that the effect is not tied to a single encoder family.

**Pass@k.** We evaluate with a $k$-way budget: *pass@k* is the number of LLM translators used in one LiRA forward pass ($k=0$ uses only MT; $k>0$ follows Table 6). Table 7 aggregates MLQA

Table 6: Translator configurations for pass@k.

| pass@k | Translator 1 | Translator 2 | Translator 3 | Translator 4 |
|---|---|---|---|---|
| pass@0 | OPUS-MT | m2m100 | nllb-200-600M | nllb-200-3.3B |
| pass@1 | Llama-3.3-70B | OPUS-MT | m2m100 | nllb-200-3.3B |
| pass@2 | Llama-3.3-70B | gemma-2-27B | m2m100 | nllb-200-3.3B |
| pass@3 | Llama-3.3-70B | gemma-2-27B | Qwen3-32B | nllb-200-3.3B |
| pass@4 | Llama-3.3-70B | gemma-2-27B | Qwen3-32B | Deepseek-R1-Distill-Qwen-32B |

Table 7: Combined pass@k results on three datasets (higher is better).

| Dataset | Method | pass@0 | pass@1 | pass@2 | pass@3 | pass@4 |
|---|---|---|---|---|---|---|
| MLQA Retrieval | Qwen3-8B | 79.96 | 80.41 | 80.45 | 80.79 | 81.13 |
| | LiRA | 81.15 | 81.56 | 81.79 | 81.53 | 81.66 |
| BelebeleRetrieval | Qwen3-8B | 82.27 | 83.54 | 83.97 | 84.55 | 85.94 |
| | LiRA | 86.00 | 86.15 | 86.67 | 86.69 | 87.03 |
| STS22 | Qwen3-8B | 69.64 | 70.01 | 70.72 | 71.32 | 71.64 |
| | LiRA | 73.01 | 73.54 | 74.11 | 74.39 | 75.00 |

Retrieval, BelebeleRetrieval, and STS22. Across $k \in \{0, 1, 2, 3, 4\}$, LiRA outperforms the baseline on all three datasets. Both models improve as $k$ increases, with diminishing gains beyond $k=2$ and a small dip at $k=3$ on MLQA. Even at $k=0$, LiRA beats the baseline, showing a tunable budget–quality trade-off without large translators at inference.

## 5.4 ABLATION

The ablation results in Table 8 demonstrate the contribution of each component in LiRA. Removing the LLM Critic or Embeds Critic leads to the most significant performance drop, particularly on Pearson correlation and accuracy, highlighting the importance of dual-level critics for effective supervision. The translation and multilingual encoder modules also provide consistent gains, showing their role in enhancing cross-lingual generalization. Finally, eliminating the FIFO loss queue results in the largest degradation on nDCG@10, confirming its necessity in stabilizing optimization. Overall, each component is essential, and their synergy ensures the robustness of LiRA across tasks.

Table 8: Ablation study of LiRA on retrieval, sentence ranking, and reasoning tasks.

| Method | nDCG@10 | Pearson | Accuracy |
|---|---|---|---|
| LiRA | **77.71** | **75.00** | **89.2** |
| - w/o LLM Critic | 71.29 | 72.19 | 87.7 |
| - w/o Embeds Critic | 65.77 | 61.78 | 81.2 |
| - w/o Translations | 75.48 | 74.39 | 86.4 |
| - w/o Multilingual encoder | 75.59 | 72.43 | 85.3 |
| - w/o FIFO loss queue | 64.29 | 69.82 | – |

## 6 CONCLUSION

We proposed LIRA, a framework for robust multilingual LLM adaptation that unifies retrieval, sentence ranking, and reasoning tasks under a common anchoring principle. By combining anchored representations with critic-guided alignment and queue-based objectives, LIRA consistently improves over strong Qwen3 baselines across both public benchmarks and our newly introduced LazRetrieval dataset. Ablation studies further validate the complementary contributions of each component. We hope our dataset and framework can inspire future work on multilingual LLM adaptation. In addition, our **Ethics Statement** and **Reproducibility Statement** are provided in the A.5 and A.6.

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

# A APPENDIX

## A.1 THEORETIC DETAILS

### A.1.1 MATH PROOF

**Theorem** (Representation deviation bound). Under Assumptions 1–2 and Definitions 1–2, let the optimal English representation be $z^\star = [\,h(y^\star);\ h(y^\star)\,]$ and the framework output be $z = [\,g(x);\ h(y)\,]$. Then

$$\|z - z^\star\|_2 \leq \epsilon_1 + C\sqrt{2\,\epsilon_2}, \tag{13}$$

where $C > 0$ is the kernel boundedness constant from Definition 1, i.e., $\sup_s k(s,s) \leq C^2$. This constant reflects the geometry of the semantic RKHS; smaller $C$ indicates more stable embeddings. In practice, $C$ can be estimated empirically on a corpus (e.g., $C \approx 0.6867$ in our experiments).

*Proof.* Let $y^\star$ be a *perfect translation* such that $p(s \mid x) = p(s \mid y^\star)$. By block structure and the triangle inequality,

$$\|\mathbf{z} - \mathbf{z}^\star\|_2 = \left\| \begin{bmatrix} g(x) - h(y^\star) \\ h(y) - h(y^\star) \end{bmatrix} \right\|_2 \leq \|g(x) - h(y^\star)\|_2 + \|h(y) - h(y^\star)\|_2. \tag{14}$$

By Assumption 1,

$$\|g(x) - h(y^\star)\|_2 \leq \|g(x) - h(y)\|_2 + \|h(y) - h(y^\star)\|_2 \leq \epsilon_1 + \|h(y) - h(y^\star)\|_2. \tag{15}$$

Next, by Assumption 2 and Pinsker's inequality,

$$\|p(s \mid y) - p(s \mid y^\star)\|_1 \leq \sqrt{2\,D_{\mathrm{KL}}\big(p(s \mid y)\,\|\,p(s \mid y^\star)\big)} \leq \sqrt{2\epsilon_2}. \tag{16}$$

Using the RKHS mean-embedding view of $h$ (Definition 1) and the bounded-kernel assumption (see, e.g., (Sriperumbudur et al., 2010)),

$$\begin{aligned}
\|h(y) - h(y^\star)\|_2 &= \big\|\mu_{p(s|y)} - \mu_{p(s|y^\star)}\big\|_{\mathcal{H}} \\
&\leq C\,\|p(\cdot \mid y) - p(\cdot \mid y^\star)\|_{\mathrm{TV}} \\
&= \frac{C}{2}\,\|p(s \mid y) - p(s \mid y^\star)\|_1 \\
&\leq C\sqrt{\tfrac{1}{2}\,D_{\mathrm{KL}}\big(p(s \mid y)\,\|\,p(s \mid y^\star)\big)} \leq C\sqrt{\tfrac{\epsilon_2}{2}}. \tag{17}
\end{aligned}$$

Plugging equation 17 into equation 15 and then into equation 14 yields

$$\left\| \mathbf{z} - \mathbf{z}^{\star} \right\|_2 \ \leq \ \left( \epsilon_1 + C \sqrt{\tfrac{\epsilon_2}{2}} \right) + C \sqrt{\tfrac{\epsilon_2}{2}} \ = \ \epsilon_1 + C \sqrt{2\,\epsilon_2} \,,$$

which proves the claim. $\square$

**Corollary.** Downstream stability. Let $f_{\mathrm{LLM}}$ denote the downstream scorer. If $f_{\mathrm{LLM}}$ is locally Lipschitz around $[g(x); h(y)]$ with constant $L^{\mathrm{loc}}(y;\delta)$ as in Definition 2, then

$$\left\| f_{\mathrm{LLM}}(\boldsymbol{z}) - f_{\mathrm{LLM}}(\boldsymbol{z}^{\star}) \right\|_2 \ \leq \ L^{\mathrm{loc}}(y;\delta) \left( \epsilon_1 + C \sqrt{2\,\epsilon_2} \right). \tag{18}$$

*Proof.* By the (local) Lipschitz property of $f_{\mathrm{LLM}}$ and Theorem 3.2,

$$\left\| f_{\mathrm{LLM}}(\boldsymbol{z}) - f_{\mathrm{LLM}}(\boldsymbol{z}^{\star}) \right\|_2 \ \leq \ L^{\mathrm{loc}}(y;\delta) \, \| \boldsymbol{z} - \boldsymbol{z}^{\star} \|_2 \ \leq \ L^{\mathrm{loc}}(y;\delta) \left( \epsilon_1 + C \sqrt{2\,\epsilon_2} \right).$$

$\square$

**Instantiation.** In our measurements we obtain $L^{(0.95)}(y;\delta) \approx 0.034$ and $C \approx 0.6867$ (representation dimension $n = 4096$). A representative bound (reported in $\ell_1$ for readability) is

$$\left\| f_{\mathrm{LLM}}(\boldsymbol{z}) - f_{\mathrm{LLM}}(\boldsymbol{z}^{\star}) \right\|_1 \ \leq \ 0.034 \cdot \left( \epsilon_1 + 1.9423 \sqrt{\epsilon_2} \right). \tag{19}$$

### A.1.2 About Definition

**Definition 1** (RKHS representation) Let $h : \mathcal{Y} \to \mathbb{R}^d$ denote the English sentence encoder. We view $h(y)$ as the *kernel mean embedding* (KME) of the conditional semantic distribution $p(s \mid y)$ in an RKHS $(\mathcal{H}, k)$:

$$h(y) \ = \ \mu_{p(s|y)} \ = \ \mathbb{E}_{s \sim p(s|y)} \big[ \varphi(s) \big], \qquad \varphi(s) = k(s, \cdot). \tag{20}$$

The kernel $k$ is assumed bounded on the (semantics) domain: $0 < k(s,s) = \langle k(s,\cdot), k(s,\cdot) \rangle_{\mathcal{H}} \leq C^2$ for some constant $C > 0$.

**Remark** (On estimating the boundedness constant $C$). For any probability measure $P$ on the input space with $x, x' \overset{\text{i.i.d.}}{\sim} P$,

$$\left\| \mu_P \right\|_{\mathcal{H}}^2 = \mathbb{E}_{x,x'}\big[ k(x, x') \big] \ \leq \ \mathbb{E}_x \big[ k(x, x) \big], \tag{21}$$

where the inequality follows from $k(x, x') \leq \sqrt{k(x,x)\,k(x',x')}$ for PSD kernels and Jensen. Thus $\|h(y)\|_{\mathcal{H}} = \|\mu_{p(s|y)}\|_{\mathcal{H}}$ provides a *lower-bound proxy* for $\mathbb{E}[k(x,x)]$, but it does not identify the *pointwise* upper bound $\sup_s k(s,s) = C^2$. In practice one may report empirical surrogates (e.g., corpus-wise maxima of $\|h(y)\|$), while the theoretical $C$ remains a kernel-dependent constant. See Smola et al. (2007); Shioda et al. (2017); Yoshikawa et al. (2015) for background.

**Estimator.** Let $E \in \mathbb{R}^{V \times d}$ be the model's input embedding table and $y = (w_1, \ldots, w_T)$ the tokenized sentence with attention mask $m_t \in \{0, 1\}$. We compute the *unnormalized* mean-pooled sentence vector

$$\widehat{h}(y) \ = \ \frac{1}{\sum_{t=1}^{T} m_t} \sum_{t=1}^{T} m_t \, E[w_t, :] \ \in \ \mathbb{R}^d, \qquad \text{and its norm} \ \ \|\widehat{h}(y)\|_2. \tag{22}$$

The corpus-level estimators are

$$\widehat{C}_{\max} \ = \ \max_{y \in \mathcal{Y}_{\mathrm{probe}}} \|\widehat{h}(y)\|_2, \qquad \widehat{C}_q \ = \ \mathrm{Quantile}_{y \in \mathcal{Y}_{\mathrm{probe}}} \big( \|\widehat{h}(y)\|_2, \, q \big), \tag{23}$$

where $q \in (0, 1)$ (e.g., $q = 0.90, 0.95, 0.99$) provides robust surrogates. By construction $\widehat{C}_{\max} \leq C$ (a *lower* bound on the true $C$).

**Implementation.** We follow the released script `compute_C_rkhs.py`: (i) tokenize each sentence, (ii) fetch token embeddings via `get_input_embeddings()`, (iii) mean-pool with the attention mask (no $\ell_2$ normalization), (iv) take $\| \cdot \|_2$ and aggregate statistics (*max*, mean, std, median, $p90$, $p95$, $p99$, sample count). Unless otherwise noted, we probe on STS22 (`sentence1`) with `max_length=8192` and report per-model results in Table 9.

Table 9: Estimates of the RKHS bound $C$ on STS22 (field: *sentence1*) using mean-pooled *unnormalized* input embeddings (no $\ell_2$ post-normalization). $\widehat{C}_{\max} = \max_y \|\widehat{h}(y)\|_2$; $\widehat{C}_q$ denotes the $q$-quantile.

| Model | $\widehat{C}_{\max}$ | Mean | Std | Median | P90 | P95 | P99 |
|---|---|---|---|---|---|---|---|
| Qwen3-Embedding-0.6B | 0.5333 | 0.1944 | 0.0221 | 0.1878 | 0.2240 | 0.2376 | 0.2605 |
| Qwen3-Embedding-4B | 0.5457 | 0.2073 | 0.0324 | 0.1971 | 0.2465 | 0.2672 | 0.3302 |
| Qwen3-Embedding-8B | 0.6866 | 0.3988 | 0.0434 | 0.3900 | 0.4529 | 0.4752 | 0.5503 |

**Analysis.** Table 9 and Figure 4 shows that the empirical RKHS bound surrogates $\widehat{C}$ increase moderately with model size (0.6B→4B→8B), which tightens separation in representation space but enlarges the worst-case radius $r(C) = \epsilon_1 + 2\,C\sqrt{2\epsilon_2}$ when plugging $C$ into our bounds. Because $\widehat{C}_{\max} \leq C$ (Definition A.1.2), any bound instantiated with $\widehat{C}_{\max}$ or $\widehat{C}_q$ is *optimistic* (it may understate the true worst case); therefore we recommend using (i) a *high-probability* bound using $C = \widehat{C}_{0.95}$ together with its empirical coverage on validation, and (ii) a *worst-case* bound using $C = \widehat{C}_{\max}$ as a lower envelope for the true $C$. For rigor, one can calibrate a multiplicative slack $\kappa \geq 1$ by back-testing—choose the smallest $\kappa$ such that the inequality with $C = \kappa\,\widehat{C}_{0.95}$ holds on at least 95% of held-out samples. Finally, $\widehat{C}$ is sensitive to tokenization length and domain; we thus compute $\widehat{C}$ on the target corpus (STS22 *sentence1* by default) with unnormalized mean pooling, and advise re-estimating it in-domain when the deployment distribution shifts.

**Definition 2** (Data-local Lipschitz constant) "How fast can the encoder's output change under small edit perturbations of a sentence in real text data?" By standard Lipschitz-continuity arguments on finite discrete domains, any encoder admits a Lipschitz constant. Hence, on the dataset $\mathcal{Y}_{\text{data}}$ the encoder satisfies

$$L_h^{\text{loc}}(y; \delta) = \max_{y' \in \mathcal{N}_\delta(y)} \frac{\left\| f_{\text{LLM}}(\mathbf{z}) - f_{\text{LLM}}(\mathbf{z}^\star) \right\|_2}{\left\| \mathbf{z} - \mathbf{z}^\star \right\|_2}, \tag{24}$$

where $\mathbf{z} = [\,g(x);\,h(y)\,]$. We denote its $q$-quantile by $L_h^{(q)}$, measured by the script described earlier (e.g., $q = 0.95$, $L_h^{(0.95)} \approx 0.05$). *Note.* $\mathcal{N}_\delta(y)$ is the neighborhood defined by token-level edit distance $\leq \delta$; in practice we use $\delta = 1$.

**Explanation.** The *data-local Lipschitz constant*

$$L_h^{\text{loc}}(y; \delta) = \max_{y' \in \mathcal{N}_\delta(y)} \frac{\left\| h(y) - h(y') \right\|_2}{d_{\text{tok}}(y, y')} \tag{25}$$

is defined as follows.

- $h : \mathcal{Y} \to \mathbb{R}^d$ is any fixed sentence encoder (e.g., Qwen-3, BERT).
- $d_{\text{tok}}(y, y')$ is the token-level edit distance between two sentences (e.g., Levenshtein distance).

**1) $\delta$-neighborhood (in the corpus).** For a finite corpus $\mathcal{Y}_{\text{data}}$ and any sentence $y$, define the radius-$\delta$ neighborhood

$$\mathcal{N}_\delta(y) = \left\{ y' \in \mathcal{Y}_{\text{data}} \ : \ 0 < d_{\text{tok}}(y, y') \leq \delta \right\}.$$

That is, $\mathcal{N}_\delta(y)$ contains all sentences that differ from $y$ by at most $\delta$ token edits (e.g., by exactly one token when $\delta = 1$).

**2) Pointwise local Lipschitz constant.** The quantity $L_h^{\text{loc}}(y; \delta)$ measures the encoder's rate of change within the data neighborhood. As long as $y$ has at least one neighbor, this value is finite.

**3) Empirical quantile over the dataset.** For a confidence level $q \in (0, 1)$ (e.g., $q = 0.95$), define

$$L_h^{(q)}(\delta) = \text{Quantile}_{y \in \mathcal{Y}_{\text{data}}} \left( L_h^{\text{loc}}(y; \delta),\, q \right). \tag{26}$$

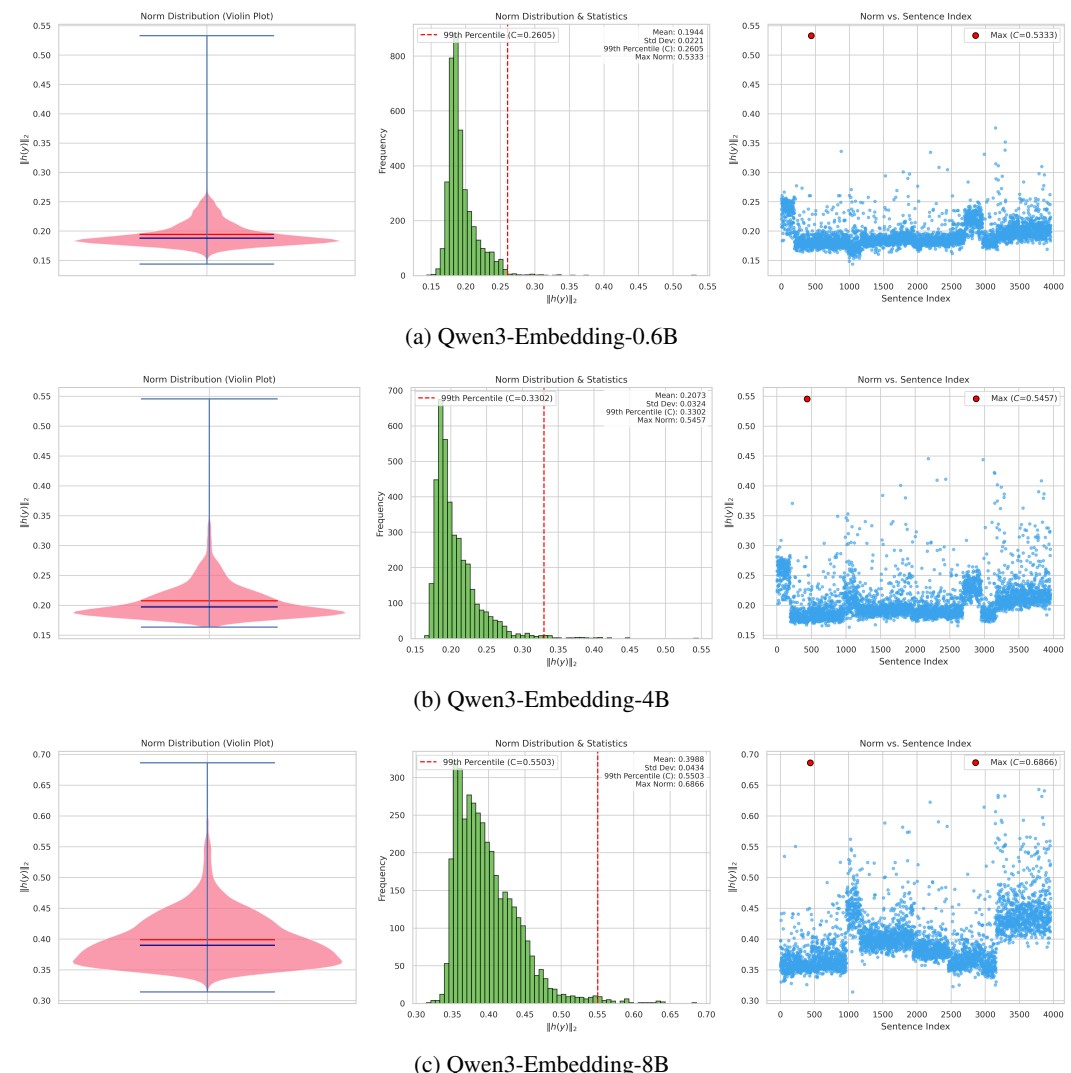

(a) Qwen3-Embedding-0.6B

(b) Qwen3-Embedding-4B

(c) Qwen3-Embedding-8B

Figure 4: Empirical estimates of the RKHS bound $C$ across model scales. Each subfigure shows (left) a violin plot of $\|\widehat{h}(y)\|_2$, (middle) a histogram with the 99th percentile marked, and (right) a scatter of norms by sentence index.

In words, $q \cdot 100\%$ of sentences in the corpus have local Lipschitz constants no greater than $L_h^{(q)}$. Empirically, for $\delta$ between 1 and 10, the local Lipschitz ratios of Qwen3-Embedding are below 0.1 for 99% of samples; moreover, as $\delta$ increases the ratios decrease and concentrate, indicating that the local Lipschitz constant both exists and is measurable.

**Experimental notes.**

- **Lipschitz Ratio ($L_h$).** For an original sentence $s$ and its perturbed version $s'$, the ratio is

$$L_h = \frac{\left\|\text{Embed}(s) - \text{Embed}(s')\right\|_2}{\text{EditDistance}(s, s')}.$$

  Here $\|\cdot\|_2$ is the Euclidean norm between embedding vectors, and $\text{EditDistance}$ is the Levenshtein distance (minimum number of single-character edits to transform $s$ into $s'$). A small and stable ratio indicates local stability/robustness: small input changes do not cause large embedding shifts. If the ratio grows markedly with $\delta$, the model may vary sharply in some regions.

Table 10: Empirical local Lipschitz estimates $L_{\text{emp}}(\delta)$ (percent). We report mean, std, median, and high quantiles (P90/P95/P99), plus max.

| Model | $\delta$ | Mean | Std | Median | P90 | P95 | P99 | Max |
|---|---|---|---|---|---|---|---|---|
| | 1 | 6.8% | 6.2% | 5% | 13.7% | 18.1% | 31.5% | 58.6% |
| | 2 | 6.5% | 5.4% | 5.1% | 12.7% | 16.3% | 27.2% | 55.7% |
| | 3 | 5.7% | 4.8% | 4.4% | 11.3% | 14.6% | 23.9% | 44.9% |
| Qwen3-Embedding-0.6B | 5 | 4.2% | 3.8% | 3.1% | 8.5% | 11.1% | 18.8% | 34% |
| | 8 | 2.7% | 3.3% | 1.7% | 5.8% | 8% | 16% | 77% |
| | 10 | 2.2% | 2.9% | 1.3% | 4.9% | 6.9% | 14.6% | 55.7% |
| | 1 | 5.5% | 5% | 4.1% | 11.1% | 14.5% | 27% | 54.9% |
| | 2 | 5.2% | 4% | 4.2% | 10% | 12.5% | 21% | 41.4% |
| | 3 | 4.6% | 3.7% | 3.7% | 8.8% | 11.3% | 18% | 38.2% |
| Qwen3-Embedding-4B | 5 | 3.4% | 3% | 2.6% | 7% | 8.7% | 14.5% | 38.7% |
| | 8 | 2.2% | 2.6% | 1.5% | 4.8% | 6.7% | 12.4% | 29.4% |
| | 10 | 1.8% | 2.4% | 1.1% | 4.1% | 5.7% | 11.9% | 39.2% |
| | 1 | 3.4% | 3% | 2.5% | 6.4% | 8.6% | 15.8% | 30.3% |
| | 2 | 3.2% | 2.6% | 2.5% | 6.1% | 7.9% | 14.3% | 29.3% |
| | 3 | 2.9% | 2.3% | 2.2% | 5.5% | 7.1% | 12.2% | 22.7% |
| Qwen3-Embedding-8B | 5 | 2.1% | 1.9% | 1.6% | 4% | 5.3% | 9.5% | 21.7% |
| | 8 | 1.4% | 1.7% | 0.9% | 2.9% | 4.1% | 8.3% | 20.4% |
| | 10 | 1.1% | 1.6% | 0.7% | 2.4% | 3.5% | 7.9% | 30% |

- $\delta$ (**Delta**). The maximum number of edit operations allowed for the perturbation. The script evaluates multiple $\delta$ values (e.g., $1, 2, 3, 5$, etc.).

- **Mean.** The average of the ratios at a fixed $\delta$, reflecting the typical sensitivity at that perturbation level.

- **Standard Deviation.** The dispersion of the ratios; larger values indicate greater variability across sentences/perturbations.

- **Quantiles.** E.g., median (50%), 90%, 95%, and 99% quantiles. The 95% quantile means that 95% of ratios are no greater than that value, useful for detecting rare but high-sensitivity cases.

- **Sample Count.** The number of valid ratios computed for a given $\delta$ (cases with no change after perturbation are excluded).

By tracking these statistics as $\delta$ varies, we assess the encoder's local Lipschitz characteristics and, in turn, its stability and robustness.

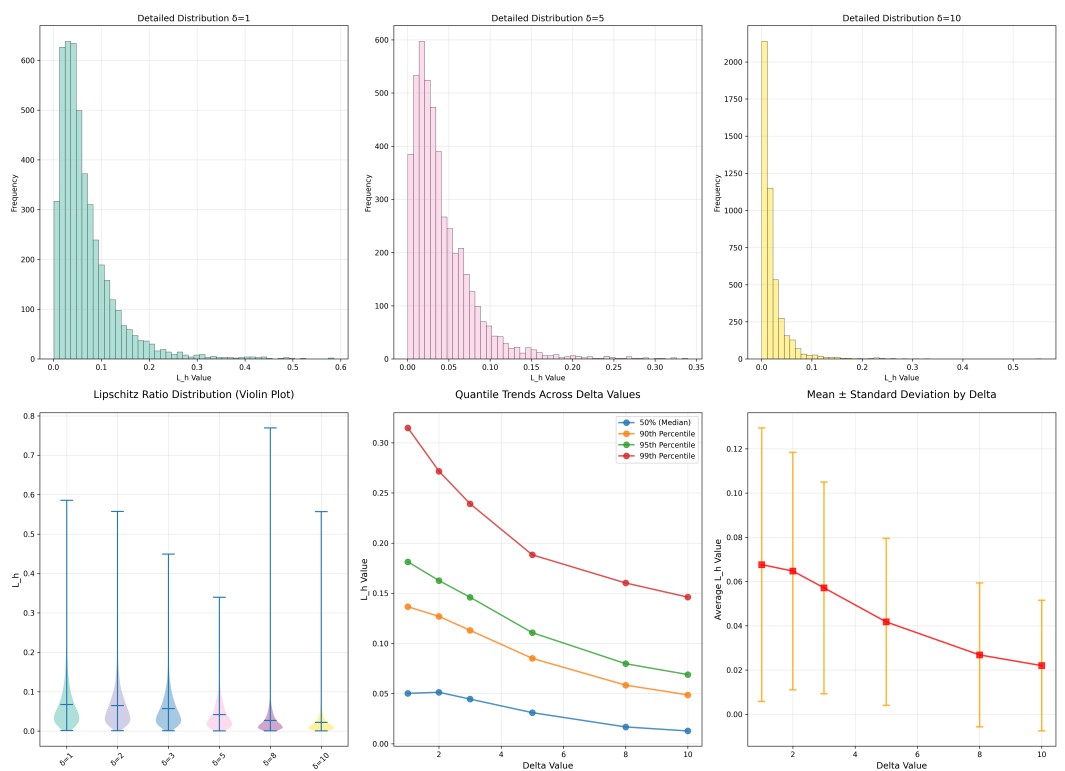

Figure 5: Qwen3-Embedding-0.6B: Local Lipschitz analysis across $\delta$.

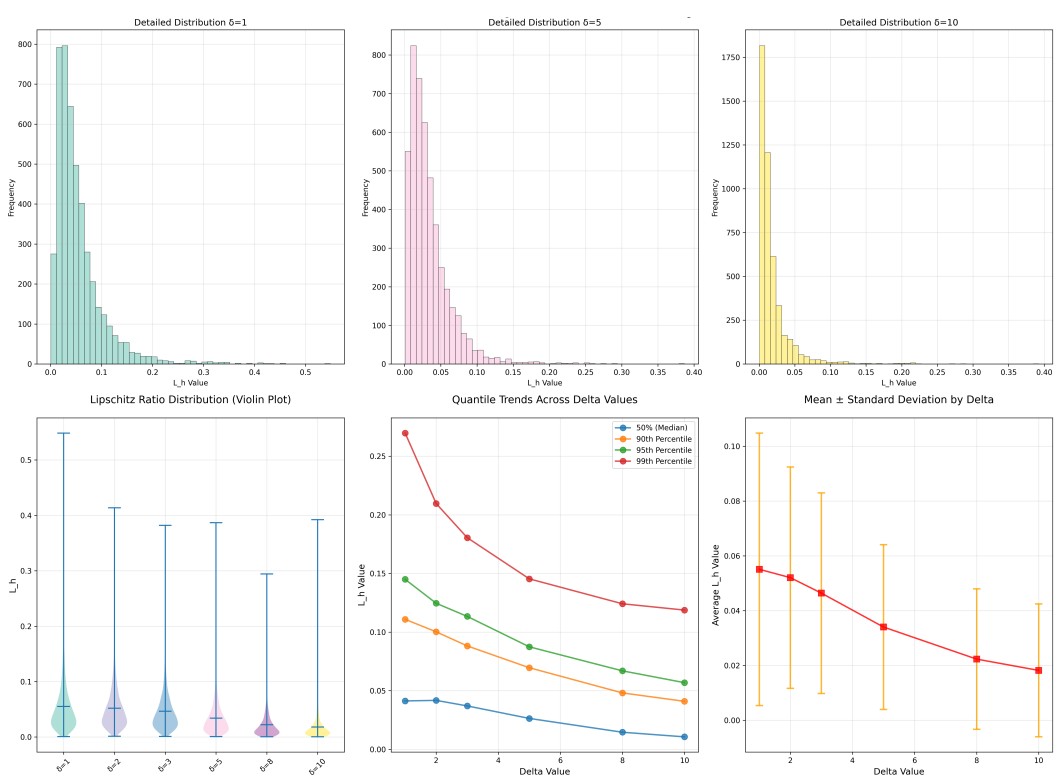

Figure 6: Qwen3-Embedding-4B: Local Lipschitz analysis across $\delta$.

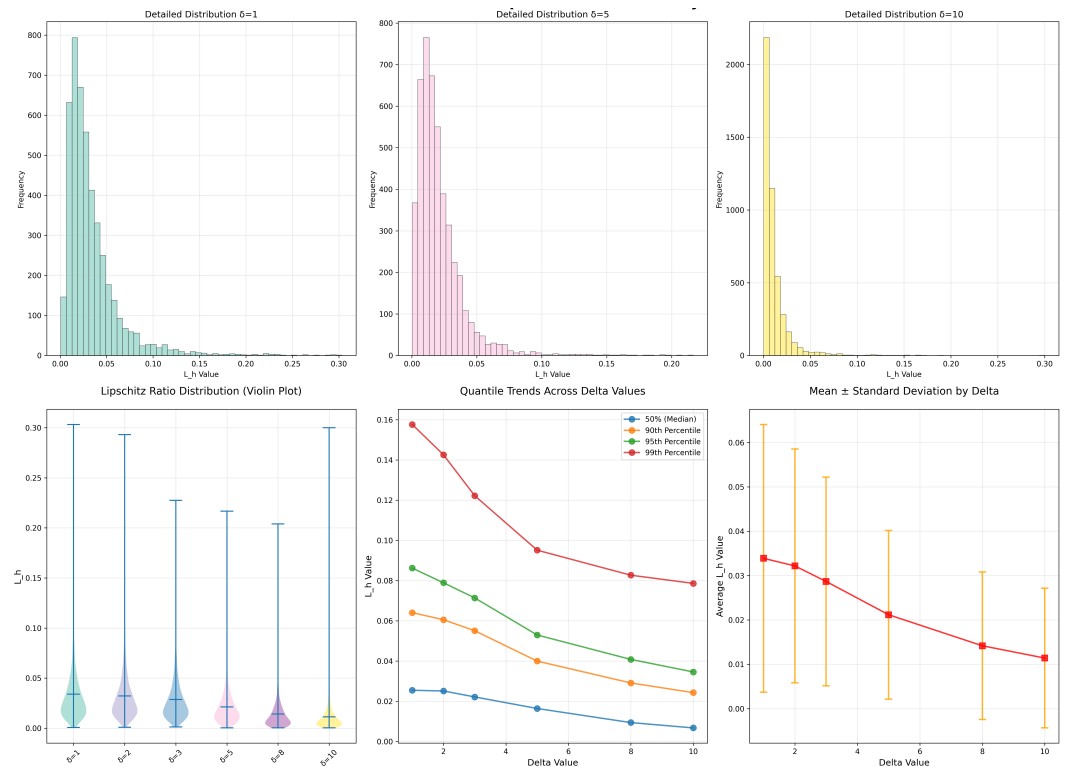

Figure 7: Qwen3-Embedding-8B: Local Lipschitz analysis across $\delta$.

**Analysis.** Across radii $\delta \in \{1, 2, 3, 5, 8, 10\}$, the empirical local Lipschitz constant $L_{\text{emp}}(\delta)$ *decreases* as $\delta$ grows, indicating smoother behavior for larger neighborhoods (finite-difference estimates are dominated by local saturation and curvature). As shown in Table 10 and Figure 5, 6, 7, scaling the encoder from 0.6B→4B→8B consistently *reduces* $L_{\text{emp}}$ at all quantiles: for example at $\delta$=5, P95 drops from 0.1107 (0.6B) to 0.0874 (4B) to 0.0530 (8B). Tail risk also shrinks (P99 and Max), with occasional outliers on 0.6B (e.g., $\delta$=8) suggesting numeric spikes; hence, for theoretical bounds we recommend using the *high-probability* constant $L_{\text{loc}} = \text{P95}$ at $\delta \in [3, 5]$ as a robust plug-in for the local bound.

## A.2 DATASET

**Dataset release.** We release a de-identified cross-lingual e-commerce retrieval dataset, **LazRetrieval**, and its pretraining-scale companion **LazRetrieval-mega**. The corpus spans seven languages across Southeast and South Asia: Vietnamese (vi), Thai (th), Indonesian (id), Malay (ms), Urdu (ur), Bengali (bn), and Filipino/Tagalog (ph). **LazRetrieval** contains $10\,\text{k}$ examples per language, while **LazRetrieval-mega** contains $1{,}000\,\text{k}$ per language. Unless otherwise specified, our experiments use *LazRetrieval*; the *mega* version is intended to support large-scale pretraining. We normalize the frequencies.

**Splits and file structure.** We split the data into train and test with a fixed 4:1 ratio. Each split consists of three JSON files:

- `query.json`: de-identified user queries from seven Lazada locales.
- `item.json`: product titles (landing-page headers) to be retrieved as candidate documents.
- `pairs_info.json`: the set of positive query–item pairs (binary relevance).

**Example.** A minimal example from the Bengali (Bangladesh) portion is shown below.

```
query:
  {
      "ID":"Q1048",
      "nation": "BD",
      "text": "সট  চুইংগাম"
  }
```

```
item:
  {
      "nation": "BD",
      "item_id": "C34801",
      "text": "ডুভেই ক্লাসিক পুরুষ ব্রেসলেট গহনা মুকুট কবজ বিলাসবহুল
ম্যাক্রাম জপমালা মহিলাদের জন্য ব্রেসলেট পালসেইরা ম্যাসকুলিনা ফেমিনিনা
উপহার"
  },
```

```
pairs_info:
  {
      "nation": "BD",
      "query_id": "Q1048",
      "query": "সট  চুইংগাম",
      "item_id": "C369",
      "item": "Trident cinnamon ফ্লেভার  সুগার  ফ্রি  গাম  x 14 সফট  গাম",
      "rscore": "1.0"
  }
```

Figure 8: Rendered examples from the Bengali (BD) split of *LazRetrieval*. We render records as images to avoid Unicode rendering issues.

Table 11: Descriptive statistics of *LazRetrieval* (train+test). Lengths are measured as raw string lengths; counts denote unique entries.

| Field | Max | Min | Mean | Median | Std | Count |
|-------|-----|-----|------|--------|-----|-------|
| Query | 248 | 2 | 18.65 | 18.00 | 9.37 | 50,000 |
| Item | 255 | 6 | 97.42 | 95.00 | 42.59 | 50,000 |

**Dataset analysis.** As shown in Table 11 and Figure 9 The corpus exhibits a pronounced length asymmetry between queries and items: queries are short on average (mean 18.66, median 18), whereas item titles are substantially longer and more dispersed (mean 97.42, std 42.59). This mismatch reflects real-world e-commerce behavior—concise user intents versus verbose product titles—and implies (i) robust handling of extreme-length outliers and (ii) sensitivity to multilingual scripts with different orthographic granularity. Our training objectives (queue-augmented correlation for sentence ranking and listwise soft-nDCG@10 with safe negatives for retrieval) are designed to be stable under such length skew, while the anchoring mechanism in LIRA mitigates representation drift caused by noisy or unusually long inputs.

## A.3 EXPERIMENTAL DETAILS

Unless stated otherwise, inputs are tokenized with `max_length=512`, `padding=true`, and truncation enabled. For consistency across backbones, document- and query-side representations are

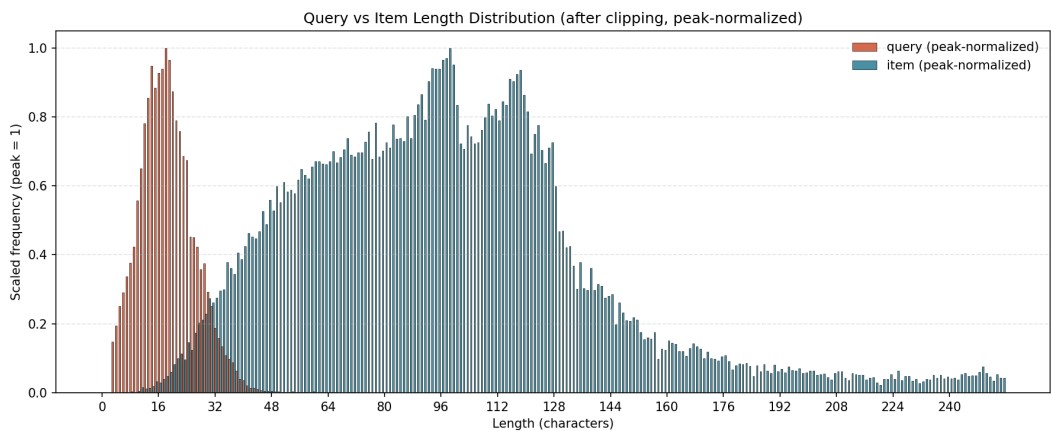

Figure 9: Length's distribution of *LazRetrieval*.

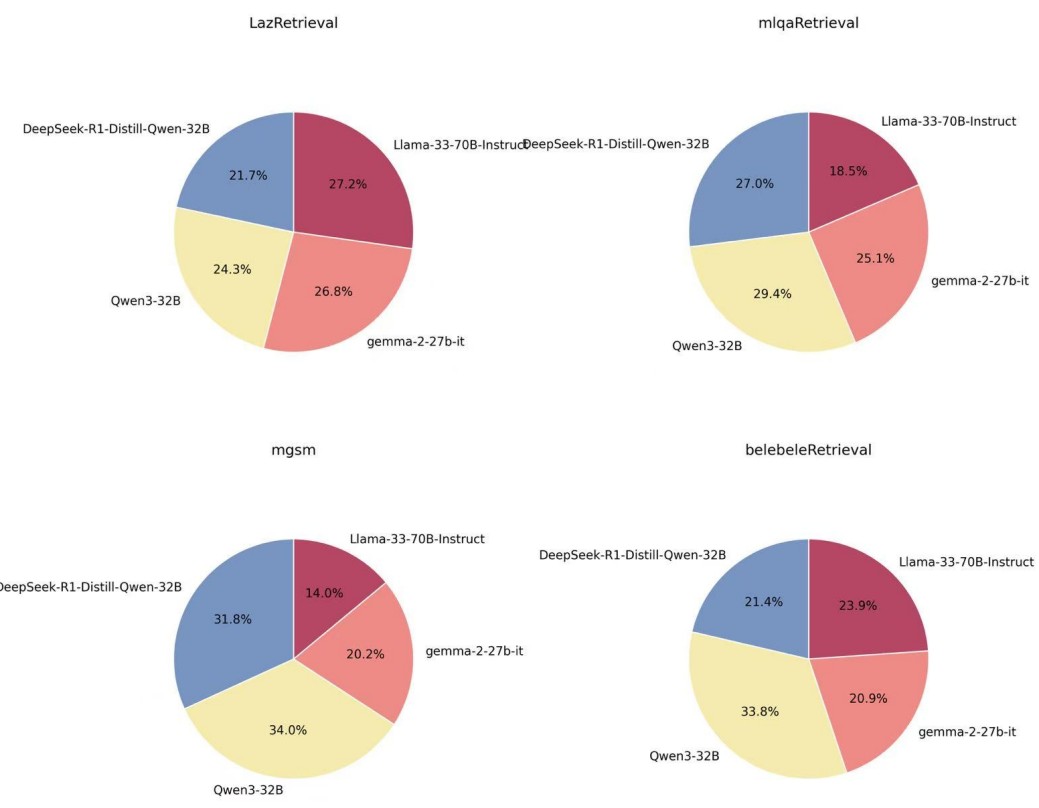

Figure 10: Model selection distribution of ARCA across four benchmarks.

pooled before similarity scoring. Training updates three learnable components: the multilingual encoder (e.g., mT5-XL encoder), the cross-space `Adaptor`, and the `Actor` selector. All are optimized with Adam and gradient clipping at 1.0. Both single-process and DDP training are supported; checkpointing and logging frequencies are configured per task (Tables. 12–13).

**Arca Translation Selections.** As shown in Figure 10. On MLQARETRIEVAL, MGSM, and BELEBELERETRIEVAL, Qwen3-32B is the most frequently selected candidate (29.4%, 34.0%, and 33.8%, respectively); together with DeepSeek-R1-Distill-Qwen-32B (27.0%, 31.8%,

Table 12: Core setup per experiment. All runs use PyTorch + Transformers; distributed training via DDP when `--distributed` is enabled.

| Task | Encoder | LLM Embedding | Pool Size | Batch | Steps | GPUs |
|------|---------|---------------|-----------|-------|-------|------|
| STS22 | mT5-XL | Qwen3-Embedding-8B | 5k | 1 | 10 | 4 |
| BelebeleRetrieval | mT5-XL | Qwen3-Embedding-8B | 1k | 1 | 5 | 8 |
| MLQARetrieval | mT5-XL | Qwen3-Embedding-8B | - | 1 | 5 | 8 |
| LazRetrieval | mT5-XL | Qwen3-Embedding-8B | 4 | 1 | 120 | 8 |
| MGSM | mT5-XL | Qwen3-4B | 4 | 2 | - | 1 |
| X-CSQA | mT5-XL | Qwen3-4B | 4 | 2 | - | 1 |

Table 13: Optimization and reward-related hyperparameters. $\alpha, \beta, \gamma$ weight the three translation scores; $\delta$ scales the similarity term. Logging and checkpoint intervals are task-specific.

| Task | Actor LR | Encoder LR | Adaptor LR | $\alpha$ | $\beta$ | $\gamma$ | $\delta$ |
|------|----------|------------|------------|----------|---------|----------|----------|
| STS22 | $1\times10^{-4}$ | $5\times10^{-5}$ | $1\times10^{-4}$ | 0.4 | 0.3 | 0.3 | 1.0 |
| BelebeleRetrieval | $1\times10^{-4}$ | $5\times10^{-5}$ | $1\times10^{-4}$ | 0.4 | 0.3 | 0.3 | 1.0 |
| MLQARetrieval | $1\times10^{-4}$ | $5\times10^{-5}$ | $1\times10^{-4}$ | 0.4 | 0.3 | 0.3 | 1.0 |
| LazRetrieval | $1\times10^{-4}$ | $5\times10^{-5}$ | $1\times10^{-4}$ | 0.4 | 0.3 | 0.3 | 1.0 |

21.4%), the Qwen-family accounts for the majority of selections (56.4%, 65.8%, and 55.2%). On LAZRETRIEVAL, the distribution is more balanced with `Llama-33-70B-Instruct` at 27.2%, `gemma-2-27b-it` at 26.8%, and the Qwen-family totaling 46.0%. Across datasets, ARCA shows a consistent preference toward Qwen-family candidates (Qwen3 or the Qwen-distilled DeepSeek variant), especially on MGSM where they comprise 65.8% of selections. A plausible explanation is architectural alignment and feature-space compatibility with our *frozen evaluator*, which is `Qwen3-72B`. Using the same family for evaluation can introduce a mild inductive bias that favors stylistic and semantic choices characteristic of that architecture. We therefore report these breakdowns to make the potential bias explicit and to encourage future work to cross-check with evaluators from different families.

**Reproducibility notes.** All models use Adam, gradient clipping (1.0), and cosine similarity on L2-normalized vectors. LLM-side token representations are extracted via `get_input_embeddings()`, and `torch.nan_to_num` is applied defensively during scoring.

### A.4 MODEL DETAILS

**Backbones. Multilingual encoder:** mT5-XL (encoder only; decoder frozen). **LLM Evaluator (frozen):** `Qwen/Qwen3-70B` is used only for evaluation/critique; all its parameters are kept frozen. **LaSR (trainable):** the LaSR module is trained end-to-end during our experiments (gradients do not propagate into the frozen evaluator). When needed, token representations are read via `get_input_embeddings()` (no gradient flow).

**Pooling & shapes.** The backbone outputs `last_hidden_state`, which is pooled to a sentence vector. On the LLM side, raw token embeddings are adaptively average-pooled to a fixed temporal length (`pool_size`, e.g., 32 or 4), then flattened for similarity computation.

**Adaptor.** A two-layer MLP maps $\mathbb{R}^{d_{\text{ML}}} \xrightarrow{\text{Linear + ReLU + LayerNorm}} \mathbb{R}^{512} \xrightarrow{\text{Linear}} \mathbb{R}^{d_{\text{LLM}}}$, aligning multilingual features to the LLM embedding space.

**Actor (candidate selector).** For each candidate, the Actor consumes a 4-D feature vector [`semantic`, `emotional`, `pragmatic`, `sim`] with topology $\mathbb{R}^4 \rightarrow 16 \rightarrow 1$ (ReLU+LayerNorm). A `softmax` over candidates defines $\pi(a \mid \mathbf{x})$, and REINFORCE is used:

$$\mathcal{L}_{\text{actor}} = -\log \pi(a \mid \mathbf{x}) \cdot R(a),$$

where $R(a) = 0.1\alpha\,\texttt{semantic} + 0.1\beta\,\texttt{emotional} + 0.1\gamma\,\texttt{pragmatic} + \delta \cdot \texttt{sim}$ (weights in Table 13).

**Encoder/adaptor objective.** We maximize cosine similarity between the selected candidate and the aligned query vector by minimizing

$$\mathcal{L}_{\text{enc}} = -\cos(\text{pool}(\text{Adaptor}(\text{Enc}_{\text{ML}}(x))),\ \text{pool}(\text{Emb}_{\text{LLM}}(y)))\,.$$

Three optimizers update `Actor`, the multilingual encoder, and the `Adaptor`; gradient clipping is applied uniformly (1.0).

**Ranking objective.** To improve the statistical stability of correlation targets (Pearson / Spearman) under small batches, we maintain a FIFO history queue of length at most $K$. At each optimization step, we concatenate the *history* (prediction–label pairs) to the current batch and compute the correlation losses jointly; the history is treated as a constant via stop-gradient and never contributes gradients. Let the current batch size be $B$, the number of valid history items be $m \le K$, and the total after concatenation be $N = m + B$. Denote the predicted similarities by $\mathbf{p} = (p_1, \ldots, p_B)^\top$ where

$$p_i = \cos(\mathbf{q}_i, \mathbf{d}_i) = \mathbf{q}_i^\top \mathbf{d}_i \in [-1, 1] \tag{27}$$

(the two vectors are $L_2$-normalized sentence embeddings), and the gold scores by $\mathbf{t} = (t_1, \ldots, t_B)^\top$ (e.g., STS22 annotations). The detached history buffers are $\mathbf{p}^{\text{hist}} \in \mathbb{R}^m$ and $\mathbf{t}^{\text{hist}} \in \mathbb{R}^m$. We concatenate

$$\tilde{\mathbf{p}} = \begin{bmatrix} \mathbf{p}^{\text{hist}} \\ \mathbf{p} \end{bmatrix}, \quad \tilde{\mathbf{t}} = \begin{bmatrix} \mathbf{t}^{\text{hist}} \\ \mathbf{t} \end{bmatrix} \in \mathbb{R}^N. \tag{28}$$

If $N < N_{\min}$ (warm-up threshold), we skip the update. *(1) Pearson correlation:*

$$r(\mathbf{a}, \mathbf{b}) = \frac{\frac{1}{N}\sum_{i=1}^{N}(a_i - \bar{a})(b_i - \bar{b})}{\sqrt{\frac{1}{N}\sum_{i=1}^{N}(a_i - \bar{a})^2 + \varepsilon}\ \sqrt{\frac{1}{N}\sum_{i=1}^{N}(b_i - \bar{b})^2 + \varepsilon}}, \quad \varepsilon = 10^{-8}, \tag{29}$$

applied to $(\tilde{\mathbf{p}}, \tilde{\mathbf{t}})$. *(2) Soft-Spearman (differentiable rank correlation):* first compute soft ranks $R_i$ for $\tilde{\mathbf{p}}$ with temperature $\tau > 0$,

$$R_i(\tilde{\mathbf{p}}; \tau) = 1 + \sum_{j=1}^{N} \sigma\left(\frac{\tilde{p}_i - \tilde{p}_j}{\tau}\right), \qquad \sigma(x) = \frac{1}{1 + e^{-x}}. \tag{30}$$

The label ranks $\rho_i(\tilde{\mathbf{t}})$ use average ties (the standard statistical convention). Define Soft-Spearman as *Pearson on ranks*:

$$r_s(\tilde{\mathbf{p}}, \tilde{\mathbf{t}}) = r\Big(\mathbf{R}(\tilde{\mathbf{p}}; \tau),\ \boldsymbol{\rho}(\tilde{\mathbf{t}})\Big). \tag{31}$$

Note that as $\tau \to 0$, $\mathbf{R}(\cdot; \tau)$ approaches discrete ranks; smaller $\tau$ sharpens sorting but increases gradient variance. *(3) Combined loss (as implemented):* for $\alpha \in [0, 1]$,

$$\mathcal{L}_{\text{CorrQ}} = \alpha\left(1 - r(\tilde{\mathbf{p}}, \tilde{\mathbf{t}})\right) + (1 - \alpha)\left(1 - r_s(\tilde{\mathbf{p}}, \tilde{\mathbf{t}})\right). \tag{32}$$

In practice, $\mathbf{p}^{\text{hist}}$ is passed via `detach()`, so gradients of $\mathcal{L}_{\text{CorrQ}}$ flow only through the current $\mathbf{p}$. The queue is updated in FIFO fashion to keep at most $K$ entries (module `corr_queue`). The warm-up threshold $N_{\min}$ (module `corr_min_effective`) enqueues without backprop when data are insufficient, stabilizing early training.

**Retrieval objective.** *Problem setup.* For each query $q$, build a candidate set $\mathcal{C} = \{d_0, \ldots, d_{C-1}\}$ where $d_0$ is the positive and the rest are online hard negatives (in-batch or mined from a same-language index). With qrels, obtain binary relevance $y_i \in \{0, 1\}$. Use inner-product/cosine scores

$$s_i = \langle \hat{q}, \hat{d}_i \rangle, \qquad \hat{q} = \frac{q}{\|q\|},\ \hat{d}_i = \frac{d_i}{\|d_i\|}. \tag{33}$$

*Differentiable ranks.* Use descending soft ranks (SoftRank) with temperature $\tau$:

$$r_i = 1 + \sum_{j \ne i} \sigma\left(\frac{s_j - s_i}{\tau}\right), \qquad \sigma(\cdot) \text{ is the sigmoid.} \tag{34}$$

Larger $s_i$ yields $r_i$ closer to $1$. *Soft nDCG@k.* Define the discount and soft top-$k$ mask as

$$\text{disc}_i = \frac{1}{\log_2(1 + r_i)}, \qquad m_i = \sigma\left(\frac{k + \frac{1}{2} - r_i}{\tau_k}\right). \tag{35}$$

With gains $g_i = 2^{y_i} - 1$,

$$\text{DCG} = \sum_i g_i \, \text{disc}_i \, m_i, \qquad \text{IDCG} = \sum_{t=1}^{k} (2^{y_t^{\downarrow}} - 1) \frac{1}{\log_2(1 + t)}, \qquad \text{nDCG@}k = \frac{\text{DCG}}{\max(\text{IDCG}, \varepsilon)}. \tag{36}$$

The base loss is

$$\mathcal{L}_{\text{ndcg}} = 1 - \text{nDCG@}k. \tag{37}$$

*Safe negatives (near-negative safety gate).* To avoid treating unlabeled relevant items as negatives, set

$$\theta = s_+ - \delta, \quad s_+ = s_0, \tag{38}$$

and identify near negatives $\{i : y_i = 0, s_i \geq \theta\}$. Apply continuous down-weighting

$$w_i = \begin{cases} \sigma\left(\dfrac{\theta - s_i}{\beta}\right), & y_i = 0 \\ 1, & y_i = 1 \end{cases} \tag{39}$$

and update $\text{disc}_i \leftarrow w_i \, \text{disc}_i$ (or drop near negatives as a more aggressive variant). *Stability terms.* To mitigate collapse and evaluation jitter, add two lightweight regularizers: (i) Top-1 hinge to enforce a margin between the positive and the hardest negative,

$$\mathcal{L}_{\text{hinge}} = \max\left(0, \ \gamma + \max_{y_i = 0} s_i - s_+\right); \tag{40}$$

(ii) Mean/variance regularization to control score centering and energy,

$$\mathcal{L}_{\text{mv}} = (\bar{s})^2 + \big| \text{Var}(s) - \nu \big|, \qquad \bar{s} = \frac{1}{C} \sum_i s_i. \tag{41}$$

*Final objective.* The per-sample objective is

$$\mathcal{L} = \mathcal{L}_{\text{ndcg}} + \lambda_h \, \mathcal{L}_{\text{hinge}} + \lambda_r \, \mathcal{L}_{\text{mv}}, \tag{42}$$

and the training loss is the batch mean. In practice we keep only in-batch negatives and take the $M$ hardest (top-$M$) to control complexity; when using external candidates (e.g., same-language queues/indices), we re-score with the current model and then take top-$M$ to reduce stale hard-negative artifacts. *Implementation notes.* We set $\tau \in [0.05, 0.2]$, $\tau_k \approx 0.5$, $\delta \in [0.1, 0.3]$, $\beta \in [0.01, 0.05]$, $\gamma \approx 0.05$, $\nu \approx 0.15$, normalize scores, apply global gradient clipping, and use EMA for evaluation. This objective directly maximizes differentiable nDCG@k while safe negatives and steady-state regularization prevent periodic collapse due to score-field saturation.

## A.5 ETHICS STATEMENT

This paper was authored by the researchers. AI tools were used only for language polishing (grammar and minor phrasing) and formatting; all ideas, methods, experiments, analyses, and conclusions are solely the authors' work.

## A.6 REPRODUCIBILITY STATEMENT

We will release our complete code—including data processing scripts and all model hyperparameters—together with the new retrieval datasets LAZRETRIEVAL and LAZRETRIEVAL-MEGA.

