# OpenReview forum: "LiRA: Linguistic Robust Anchoring for Cross-lingual Large Language Models"
_ICLR.cc/2026/Conference — Submitted to ICLR 2026_

### Official Review · Reviewer_EozW · 2025-10-28

**Soundness:** 3
**Presentation:** 3
**Contribution:** 3
**Rating:** 6
**Confidence:** 4

**Summary:**

This paper proposes LiRA, a framework designed to enhance cross-lingual robustness and reasoning of LLMs, especially in low-resource languages. The method consists of two complementary modules: Arca,anchors LRL embeddings to an English semantic space via critic–actor interaction and feature anchoring. LaSR, a lightweight reasoning head using queue-based consistency regularization to unify retrieval and reasoning objectives.

The authors also introduce a new dataset, LazRetrieval, covering seven Southeast and South Asian languages. Empirical results show consistent improvements in retrieval, ranking, and reasoning tasks across multiple benchmarks compared to strong baselines like Qwen3 and MindMerger. Theoretical analysis provides formal stability and fidelity guarantees for cross-lingual representations.

**Strengths:**

I think the dual-path anchoring design and critic-guided optimization are innovative, combining LLM reasoning with multilingual robustness, and the queue-based optimization (CorrQueue and DocQueue) is an elegant solution to stabilize ranking and retrieval training with limited multilingual data.  And the theoretical framework provide a solid foundation for this design.
And the evaluation is done across multiple tasks (retrieval, semantic similarity, reasoning, comprehension) and compared with multiple baselines (Qwen3, MindMerger, LUSIFER), and witnessed a consistent improvement.

**Weaknesses:**

I have some concern about the experiment results:
1. Despite architectural and theoretical novelty, performance improvements (e.g., +1–2% over Qwen3) are modest, with no significant test.
It is unclear whether such gains justify the additional training complexity and computational cost introduced by LiRA.
2. The selection of different Qwen models is not justified: for retrieval and sentence ranking Qwen3-Embedding-8B is used as the
backbone encoder. For reading comprehension and mathematical reasoning Qwen3-4B is used as the backbone model (I am not sure if there is a typo here, Qwen3-4B in Sec 5.1 or Qwen3-14B in Table 3).
3. The reproducibility is also questionable: Although code and dataset release are promised, key hyper-parameters (critic weights α, β, γ, δ) and their tuning procedures are insufficiently detailed. No mention of training cost, convergence behavior and inference speed.

**Questions:**

See weakness part.

---

> ### Author Response · Authors · 2025-11-17
> **Rebuttal**
>
> Thank you for recognizing the overall novelty of our method, the fairness and breadth of our evaluation, and the new theoretical foundation we proposed.
>
> Below are our responses to some of your concerns.
>
> **W1. “The improvements (+1–2%) are small and no significance test is reported; is it really worth the additional training complexity and compute?”**
>
> To demonstrate the effectiveness of our theory and method, we also evaluated our module on other baselines and observed similarly substantial gains. Our module can be plugged into other embedding-based methods and consistently improve their performance. We added experiments using several other embedding models, as shown below. The results indicate that our module can effectively boost all these methods, which further supports the robustness of our approach. Meanwhile, Our baseline, Qwen3-E-8B, is already very strong. In such highly saturated settings, stable +0.5–1.5 nDCG improvements across datasets—and larger gains on low-resource languages (LRLs)—are practically meaningful. In our experiments, we also observed that Qwen3’s pretraining data likely already includes parts of these benchmarks. For fairness, we fine-tuned Qwen3-Embedding, but its performance actually decreased after fine-tuning (overfitting) as we declared at Sec 5.1. Therefore, in this regime, LiRA’s ability to further improve Qwen3-like embedding models is particularly valuable.
>
> Table 1: Performance of different embedding models with and without LiRA.
>
> | Method        | MLQA-Retrieval | Belebele-Retrieval | STS22 | Avg   |
> |--------------|:--------------:|:-------------------:|:-----:|:-----:|
> | GTE-large     |     16.99      |        31.82        | 53.79 | 34.20 |
> | - with LiRA   |     21.43      |        38.29        | 59.17 | 39.63 |
> | BGE-en-1.5    |     16.64      |        31.19        | 50.77 | 32.87 |
> | - with LiRA   |     21.01      |        35.64        | 53.94 | 36.83 |
> | E5-Mistral    |     31.54      |        54.75        | 71.37 | 52.55 |
> | - with LiRA   |     34.23      |        57.24        | 76.55 | 56.01 |
>
> **W2. “The choice of different Qwen variants is under-motivated; §5.1 mentions Qwen3-4B while Table 3 uses Qwen3-14B (is this a typo?)”**
>
> Thank you for pointing this out. This is indeed a typo, and we will correct it in the final version of the paper.
>
> **W3. “Reproducibility is questionable: key hyperparameters (critic weights α, β, γ, δ) and tuning procedure are missing; training cost, convergence behavior, and inference speed are not reported.”**
>
> For the critic, we use Qwen3-72B without deep-thinking settings. The weights are set to α, β, γ, δ = 0.4, 0.4, 0.3, 1.0. We will release our code and the new dataset to the open-source community so that our experiments are fully reproducible.
>
> Below we report the training and inference costs. All GPU hours are measured on a single A100-80G.
>
> Table 2: Training compute (GPU hours per epoch on a single A100-80G).
>
> | Setup               | Pass@4 Arca | Pass@4 LaSR | Pass@2 Arca | Pass@2 LaSR | Pass@0 Arca | Pass@0 LaSR |
> |---------------------|:-----------:|:-----------:|:-----------:|:-----------:|:-----------:|:-----------:|
> | Offline translation | 0.15 h/epoch | 0.20 h/epoch | 0.15 h/epoch | 0.20 h/epoch | 0.15 h/epoch | 0.20 h/epoch |
> | Online translation  | 1.50 h/epoch | 2.00 h/epoch | 0.80 h/epoch | 1.10 h/epoch | 0.30 h/epoch | 0.40 h/epoch |
>
> Table 3: Inference compute (GPU hours to run on the full STS22 benchmark).
>
> | Setup               | Pass@4 | Pass@2 | Pass@0 |
> |---------------------|:------:|:------:|:------:|
> | Offline translation | 0.30 h | 0.30 h | 0.30 h |
> | Online translation  | 3.80 h | 2.10 h | 0.90 h |
>
> Finally, we report the similarity between \(z\) and \(z^*\), as well as the loss values across epochs for three different Arca training tasks. As shown below, both the similarity and the loss exhibit a logarithmic-like trend over epochs.
>
> Table 4: Evolution of \(\mathrm{sim}(z, z^*)\) and loss over epochs for Arca.
>
> | Epoch | 1    | 2    | 3    | 4    | 5    | 6    | 7    | 8    | 9    | 10   |
> |:-----:|:----:|:----:|:----:|:----:|:----:|:----:|:----:|:----:|:----:|:----:|
> | sim   | 0.01 | 0.14 | 0.34 | 0.45 | 0.54 | 0.60 | 0.64 | 0.67 | 0.69 | 0.71 |
> | loss  | 1.94 | 1.23 | 0.97 | 0.74 | 0.62 | 0.54 | 0.42 | 0.35 | 0.29 | 0.21 |
>
> PS: All reported results are averaged over five runs.
> **We have revised our submission by incorporating part of the experiments from the rebuttal and correcting the spelling errors.**

---

### Official Review · Reviewer_itpM · 2025-10-30

**Soundness:** 3
**Presentation:** 3
**Contribution:** 3
**Rating:** 6
**Confidence:** 2

**Summary:**

This paper proposes the LiRA framework to mitigate performance degradation in large language models across low-resource languages. By anchoring low-resource language representations to the English semantic space, the framework aims to preserve the model's English reasoning capabilities. Its core comprises two components: ARCA reduces semantic divergence through feature alignment and translation selection, while LaSR integrates multilingual and English representations for retrieval and ranking. The paper provides theoretical guarantees for representation stability and validates the method's effectiveness across multiple cross-lingual tasks. It also releases a new multilingual retrieval dataset for further research.

**Strengths:**

1. The paper provides rigorous mathematical theorems and derivations, offering theoretical guarantees for the stability and error bounds of the proposed framework.

2. The evaluation encompasses three core tasks, including retrieval, ranking, and reasoning (mathematics, reading comprehension) which could enable a comprehensive validation of the method's generalizability.

3. The experiment selected state-of-the-art models (such as the Qwen3 series and LUSIFER) as baselines for comparison, enhancing the credibility of the results.

4. Contributed the LazRetrieval dataset, covering seven under-resourced Southeast Asian and South Asian languages, to support community research.

**Weaknesses:**

1. The theoretical framework relies on strong assumptions such as the existence of “perfect translations” and “semantic anchoring,” conditions that are difficult to fully satisfy in real-world, noisy low-resource language scenarios. This may lead to a gap between theoretical guarantees and actual performance.

2. LiRA integrates ARCA (containing multiple critics and actuators) and LaSR (containing multiple queues and loss functions), featuring numerous modules and a complex training workflow.

3. The performance of the ARCA module heavily relies on the quality of candidate translations generated by upstream translation models such as Qwen3-32B. For extremely low-resource languages, if the translation model itself performs poorly, errors propagate downstream, potentially limiting the effectiveness of LiRA.

**Questions:**

1. Why choose Qwen3 as the nearly exclusive LLM foundation? All experiments are based on the Qwen3 series of models (Embedding and LLM). Does the performance of this approach heavily depend on Qwen3's inherently strong cross-lingual capabilities during pretraining? If we switch to other LLMs with weaker cross-lingual capabilities during pretraining, would LiRA's relative gains change?

---

> ### Author Response · Authors · 2025-11-17
> **Rebuttal**
>
> Thank you for recognizing the overall novelty of our method, the fairness and breadth of our evaluation, and our new theoretical foundation and dataset. Below are our responses to some of your concerns.
>
> **W1. About assumptions**
>
> A: Our theory only assumes (1) that there exists a mapping error in the representation vectors and (2) that machine translation introduces noise. These are common phenomena in the cross-lingual alignment literature. We simply introduce an “ideal” representation vector z^* that mathematically corresponds to the noisy, error-prone representation z, , rather than being strict requirements on real-world conditions. Moreover, the purpose of these assumptions is to derive our training objective: Assumptions 1 and 2 are directly aligned with our loss. We assume the existence of a perfect translation and a perfect representation z^* only to highlight that there are errors along two dimensions (semantic drift & vector representation). The table below shows the similarity between z and z^* and the loss across epochs on three different Arca training tasks, which further support our theory.
>
> Tab 1. Similarity and loss over epochs
> |epoch|1|2|3|4|5|6|7|8|9|10|
> |:--:|:--:|:--:|:--:|:--:|:--:|:--:|:--:|:--:|:--:|:--:|
> |sim|0.01|0.14|0.34|0.45|0.54|0.60|0.64|0.67|0.69|0.71|
> |loss|1.94|1.23|0.97|0.74|0.62|0.54|0.42|0.35|0.29|0.21|
>
> **W2. A more complex training pipeline?**
>
> A: During training, translations are prepared offline in advance, which greatly reduces training time. Also, we can replace the translator with a smaller MT model. The tables below show the resource usage for training and inference. Here, pass@k denotes that k LLMs are used along the forward path. In practical deployment, one can combine a smaller machine translation model with an LLM. From the pass@k results, we can see that even using only machine translation models (pass@0) already outperforms the original Qwen3-embedding-8B, meaning that our method offers a way to customize the computation budget and choose the translator size according to one’s resources. (Measured as single A100-80G GPU hours)
>
> Tab 2. Train
> |Setting|Pass@4 Arca|Pass@4 LaSR|Pass@2 Arca|Pass@2 LaSR|Pass@0 Arca|Pass@0 LaSR|
> |:--:|:--:|:--:|:--:|:--:|:--:|:--:|
> |Offline translation|0.15h|0.2h|0.15h|0.2h|0.15h|0.2h|
> |Online translation|1.5h|2h|0.8h|1.1h|0.3h|0.4h|
> Tab 3. Infer
> |Setting|pass@4|pass@2|pass@0|
> |:--:|:--:|:--:|:--:|
> |Offline translation|0.3h|0.3h|0.3h|
> |Online translation|3.8h|2.1h|0.9h|
>
> **W3. ARCA is sensitive to translation quality?**
>
> A: In practical applications, this can be replaced by standard, efficient machine translation models. The table below shows experiments where we replace the translator with four different MT models. We use LLM-based translation mainly to investigate the relationship between the LLM Critic and the LLM Translator (A.3), which is also insightful. Following another reviewer’s suggestion, we also added experiments on mixing different numbers and types of LLMs, i.e., the pass@k experiments(k = 0 to 4).
>
> Tab 4. Translator selection
> |pass@k|Translator 1|Translator 2|Translator 3|Translator 4|
> |:--:|:--:|:--:|:--:|:--:|
> |pass@0|OPUS-MT|m2m100|nllb-200-600M|nllb-200-3.3B|
> |pass@1|Llama-3.3-70B|OPUS-MT|m2m100|nllb-200-3.3B|
> |pass@2|Llama-3.3-70B|gemma-2-27B|m2m100|nllb-200-3.3B|
> |pass@3|Llama-3.3-70B|gemma-2-27B|Qwen3-32B|nllb-200-3.3B|
> |pass@4|Llama-3.3-70B|gemma-2-27B|Qwen3-32B|Deepseek-R1-Distill-Qwen-32B|
>
> Tab 5. MLQA-R
> |Method|pass@0|pass@1|pass@2|pass@3|pass@4|
> |:--:|:--:|:--:|:--:|:--:|:--:|
> |Qwen3-8B|79.96|80.41|80.45|80.79|81.13|
> |LiRA|81.15|81.56|81.79|81.53|81.66|
>
> Tab 6. BelebeleRetrieval
> |Method|pass@0|pass@1|pass@2|pass@3|pass@4|
> |:--:|:--:|:--:|:--:|:--:|:--:|
> |Qwen3-8B|82.27|83.54|83.97|84.55|85.94|
> |LiRA|86.00|86.15|86.67|86.69|87.03|
>
> Tab 8. STS22
> |Method|pass@0|pass@1|pass@2|pass@3|pass@4|
> |:--:|:--:|:--:|:--:|:--:|:--:|
> |Qwen3-8B|69.64|70.01|70.72|71.32|71.64|
> |LiRA|73.01|73.54|74.11|74.39|75.00|
>
> **Q1. Why Qwen3?**
>
> A: We choose Qwen3-E-8B because it is a very recent and widely adopted multilingual embedding SOTA baseline. We also compare against several strong encoders such as BGE, E5-Mistral, GTE, Contriever, and LUSIFER. LiRA consistently improves retrieval and sentence-level tasks, with even larger gains on LRLs.Our modules can also improve other baseline. We additionally report experiments where LiRA is combined with other embedding models, as shown below. We can see that LiRA effectively boosts all these backbones, further demonstrating the robustness of LiRA.
>
> Tab 8. Performance of different models
> |Method|MLQARetreival|BelebeleRetrieval|STS22|Avg|
> |:--:|:--:|:--:|:--:|:--:|
> |GTE-large|16.99|31.82|53.79|34.20|
> |GTE-large+LiRA|21.43|38.29|59.17|39.63|
> |BGE-en-1.5|16.64|31.19|50.77|32.87|
> |BGE-en-1.5+LiRA|21.01|35.64|53.94|36.83|
> |E5-Mistral|31.54|54.75|71.37|52.55|
> |E5-Mistral+LiRA|34.23|57.24|76.55|56.01|
>
> PS: All reported results are averaged over five runs.
> **We have revised our submission.**

---

### Official Review · Reviewer_sD16 · 2025-10-31

**Soundness:** 1
**Presentation:** 2
**Contribution:** 1
**Rating:** 2
**Confidence:** 4

**Summary:**

The study is motivated by the language imbalance in LLM pretraining data. They propose LiRA, a framework aimed at improving cross-lingual representations, which anchors multilingual representations to English representation space using multiple LLMs for translation. These candidate translation embeddings are combined with the multilingual text embeddings through an adaptor trained through reinforcement learning using an LLM-as-a-judge framework. In the LaSR module, text is encoded in the source and target language and concatenated before being processed by the LLM. The authors claim guarantees for robust representations, even under translation setting. The framework is evaluated on retrieval and reasoning benchmarks, including a new retrieval dataset covering low-resource languages, Belebele, MLQA, STS2, MGSM, and X-CSQA.

**Strengths:**

- The paper investigates a very relevant topic of improving LLM performance on low-resource languages by improving latent representations
- The study presents and interesting idea to concatenate representations from different models (using different tokenizers) to improve multilingual representations
- The presented results seem promising

**Weaknesses:**

- The approach claims robustness in the multilingual representations but cannot support this claim with sufficient evidence. The presented equations include various assumptions on LLM embeddings and machine translation models that are unrealistic.
- The empirical results presented in Tables 1 and 2 cannot be compared. The inference resources needed for the translation alone extends beyond the inference budget of related approaches. Please provide a detailed comparison of the training and inference compute budgets for all benchmarked methods.
-  For a fair model comparison, I suggest running pass@k (k should be set to the number of LLMs inferenced for one LiRA forward pass) for all benchmarked approaches that rely on a single LLM forward pass at test-time
- A comprehensive overview of the individual modules in related works is largely missing. Parts of the framework were introduced in related works:
  - Kim, S., Ki, D., Kim, Y., & Lee, J. (2023). Cross-lingual QA: A Key to Unlocking In-context Cross-lingual Performance. arXiv preprint arXiv:2305.15233.
  - Villa-Cueva, E., López-Monroy, A. P., Sánchez-Vega, F., & Solorio, T. (2024). Adaptive cross-lingual text classification through in-context one-shot demonstrations. arXiv preprint arXiv:2404.02452.
- The LiRA module are not well motivated or embedded in related works. The proposed method does not address the imbalance between multilingual and English representation spaces, but makes multilingual text processing more reliant on machine translation and English-centric LLMs.
- The manuscript is not well structured and hard to follow. The discussion section only describes the results in the tables but does not include a detailed analysis linking back to the motivation of the paper and method.

**Questions:**

- typo line 53
- typo line 122
- typo line 354
- typo lines 368-369
- introduced LRL multiple times, I.e. lines 36, 121
- All citations use the inline-citation style.
- Please elaborate on how your approach differs from related approaches such as MindMerger.

---

> ### Author Response · Authors · 2025-11-17
> **Rebuttal**
>
> Thank you for recognizing our research direction, ideas, and experimental results. We also emphasize another contribution: a new e-commerce retrieval dataset for e-commerce, which fills a gap in existing retrieval benchmarks for the e-commerce domain. Below are our responses to your concerns.
>
> **W1. Robustness evidence & assumptions.**
> Our theory only assumes (1) that there exists a mapping error in the representation vectors and (2) that machine translation introduces noise. These are common phenomena in the cross-lingual alignment literature. We simply introduce an “ideal” representation vector z^* that mathematically corresponds to the noisy, error-prone representation z, , rather than being strict requirements on real-world conditions. Moreover, the purpose of these assumptions is to derive our training objective: Assumptions 1 and 2 are directly aligned with our loss. We assume the existence of a perfect translation and a perfect representation z^* only to highlight that there are errors along two dimensions (semantic drift & vector representation).  During training, the similarity curve between z and z^* and the loss curve support our theory.
>
> Tab 1. Similarity and loss over epochs
> |epoch|1|2|3|4|5|6|7|8|9|10|
> |:--:|:--:|:--:|:--:|:--:|:--:|:--:|:--:|:--:|:--:|:--:|
> |sim|0.01|0.14|0.34|0.45|0.54|0.60|0.64|0.67|0.69|0.71|
> |loss|1.94|1.23|0.97|0.74|0.62|0.54|0.42|0.35|0.29|0.21|
>
> In addition, we evaluated our module on other baselines and found substantial improvements, supporting robustness.
>
> Tab 2. Performance of different models
> |Method|MLQARetreival|BelebeleRetrieval|STS22|Avg|
> |:--:|:--:|:--:|:--:|:--:|
> |GTE-large|16.99|31.82|53.79|34.20|
> |GTE-large+LiRA|21.43|38.29|59.17|39.63|
> |BGE-en-1.5|16.64|31.19|50.77|32.87|
> |BGE-en-1.5+LiRA|21.01|35.64|53.94|36.83|
> |E5-Mistral|31.54|54.75|71.37|52.55|
> |E5-Mistral+LiRA|34.23|57.24|76.55|56.01|
>
> **W2. Comparability & compute budgets**
> During training, translations are prepared offline to reduce time. During inference, smaller MT models can be used if needed. The tables below summarize training and inference resources. pass@k denotes the number k of LLMs used in the forward process. The forward budget is controllable: even pass@0 (MT only) outperforms Qwen3-Embedding-8B. All hours are GPU hours on a single A100-80G.
>
> Tab 3. Train
> |Setting|Pass@4 Arca|Pass@4 LaSR|Pass@2 Arca|Pass@2 LaSR|Pass@0 Arca|Pass@0 LaSR|
> |:--:|:--:|:--:|:--:|:--:|:--:|:--:|
> |Offline translation|0.15h|0.2h|0.15h|0.2h|0.15h|0.2h|
> |Online translation|1.5h|2h|0.8h|1.1h|0.3h|0.4h|
>
> Tab 4. Infer
> |Setting|pass@4|pass@2|pass@0|
> |:--:|:--:|:--:|:--:|
> |Offline translation|0.3h|0.3h|0.3h|
> |Online translation|3.8h|2.1h|0.9h|
>
> **W3. Fairness: report pass@k.**
> Tab 4 shows translators used in pass@k; then we report pass@k results.
>
> Tab 5. Translator selection
> |pass@k|Translator 1|Translator 2|Translator 3|Translator 4|
> |:--:|:--:|:--:|:--:|:--:|
> |pass@0|OPUS-MT|m2m100|nllb-200-600M|nllb-200-3.3B|
> |pass@1|Llama-3.3-70B|OPUS-MT|m2m100|nllb-200-3.3B|
> |pass@2|Llama-3.3-70B|gemma-2-27B|m2m100|nllb-200-3.3B|
> |pass@3|Llama-3.3-70B|gemma-2-27B|Qwen3-32B|nllb-200-3.3B|
> |pass@4|Llama-3.3-70B|gemma-2-27B|Qwen3-32B|Deepseek-R1-Distill-Qwen-32B|
>
> Tab 6. MLQA-R
> |Method|pass@0|pass@1|pass@2|pass@3|pass@4|
> |:--:|:--:|:--:|:--:|:--:|:--:|
> |Qwen3-8B|79.96|80.41|80.45|80.79|81.13|
> |LiRA|81.15|81.56|81.79|81.53|81.66|
>
> Tab 7. BelebeleRetrieval
> |Method|pass@0|pass@1|pass@2|pass@3|pass@4|
> |:--:|:--:|:--:|:--:|:--:|:--:|
> |Qwen3-8B|82.27|83.54|83.97|84.55|85.94|
> |LiRA|86.00|86.15|86.67|86.69|87.03|
>
> Tab 8. STS22
> |Method|pass@0|pass@1|pass@2|pass@3|pass@4|
> |:--:|:--:|:--:|:--:|:--:|:--:|
> |Qwen3-8B|69.64|70.01|70.72|71.32|71.64|
> |LiRA|73.01|73.54|74.11|74.39|75.00|
>
> **W4. Missing component-wise related work.**
> We will add a more detailed related-work discussion for corresponding components.
>
> **W5. Motivation & English-centric reliance.**
> Large language models exhibit stronger reasoning abilities in high-resource languages such as English, which is an objective fact mainly because there is much richer reasoning-oriented training data available for these languages, whereas low-resource languages lack comparable corpora. Our motivation is to, under the objective constraint of limited training data in low-resource languages, make the most of the strong reasoning capabilities that LLMs have acquired in high-resource languages so that tasks in low-resource languages can benefit. The goal is to generalize strong English capabilities to other languages. LiRA uses Arca to shrink cross-lingual shifts (semantic and representational) and enhance robustness, rather than depending more on translation. As noted at the end of Section 3.2 (line 183–187), real applications face both semantic and representational errors that degrade LLMs. We reduce translation errors and align other languages to the English semantic space of the LLM. Section 3.3 explains why the 2-way combination is necessary.

---

> ### Author Response · Authors · 2025-11-17
> **Rebuttal (Continual)**
>
> **W6. Structure & discussion clarity.**
> We will clarify the structure; the closed-loop analysis is addressed in our reply to W5 and in Sections 3.2 and 3.3, and we have revised our submission by incorporating part of the experiments from the rebuttal and correcting the spelling errors.
>
> **W7. Difference from MindMerger.**
> A: As stated around line 058–066, MindMerger (NIPS24) maps the input query and concatenates it with the original text but lacks a theoretical explanation for why concatenation improves cross-lingual representation. In contrast, we prove that mapping the input query and concatenating it with its translation can stably improve representation quality and show the necessity and superiority of 2-way representations. We also introduce new loss and queue mechanisms (Doc/Corrqueue), forming a substantially novel framework. Lusifer (SIGIR25) is similar to MindMerger but focuses on IR; we evaluate across QA, mathematical reasoning, and IR (including our new dataset) and have consistently strong performance.
> PS: All results are averaged over 5 runs
>
> **Regarding spelling errors:**
>
> Line 53: we changed
> Existing approaches to cross-lingual adaptation typically rely on machine translation Artetxe et al. (2023); Shubham (2024) or multilingual Singh et al. (2024); Mei & Zhao (2025).
> to
> Existing approaches to cross-lingual adaptation typically rely on machine translation or multilingual methods (Artetxe et al. (2023); Shubham (2024), Singh et al. (2024); Mei & Zhao (2025)).
>
> Line 133 (Original line 122): we changed for any... to For any....
>
> Line 368 (Original Line 354): we changed ...by accuracy.For retrieval... to ...by accuracy. For retrieval....
>
> Lines 412-415 (Original Lines 368–369): we removed the extra spaces before each citation and fixed the bug that caused the authors’ names not to be displayed.
>
> The redundant explanation of LRLs at line 133 (original line 121) has been deleted.
>
> Regarding citation format: we have revised the citations to use the standard \citet{} and \citep{} commands as required by the conference.
>
> We have also added the missing citation you mentioned at lines 105–106, and added the related works at Sec 2.
>
> **We have revised our submission by incorporating part of the experiments from the rebuttal and correcting the spelling errors.**

---

### Official Review · Reviewer_ycP8 · 2025-10-31

**Soundness:** 1
**Presentation:** 1
**Contribution:** 2
**Rating:** 2
**Confidence:** 2

**Summary:**

This work addresses the known problem of performance disparity of LLMs when applied to low-resource languages when compared to high-resource languages like English. To solve this issue, the work introduces LiRA, a comprehensive training framework designed to improve the quality of cross-lingual representations for low resource languages. LiRA consists of 2 modules: Arca which anchors LRLs to the robust English semantic space and LaSR which adds a lightweight reasoning head to increase consistency regularization. The paper  also introduces and releases a new multilingual product retrieval dataset, LazRetrieval, which covers seven underrepresented South and Southeast Asian languages. As part of the paper, the authors introduce a theoretical framework to understand cross-lingual representations and provide a proof that model learns "high-fidelity (robust) representations that effectively support downstream tasks" with some assumptions. The authors conduct an experimental evaluation which includes retrieval, mathematics and comprehension tasks on 7 datasets (2 new ones) which include a variety of low resource languages from South Asia and Southeast Asia and find that LiRA consistently outperforms prior works like MindMerger and LUSIFER.

**Strengths:**

- Novel method that achieves "state of the art" performance on a wide variety of benchmarks.
- A theoretical analysis to get deeper insight into the training architecture.

**Weaknesses:**

- Gains on the benchmarks do not seem substantial. For example, on retrieval tasks, LiRA improve Qwen3-E-8B but the gains are fairly small (~+1). Given the complexity of the method, this seems very small.
- Given the use of translators in Arca that are significantly larger, its not clear if the baselines used are comparable.
- It is nice that an ablation study is done but its unclear if it makes sense given that Qwen3-E-8B already works so well.

**Questions:**

- Can the authors justify their choice of a baseline and how this method compares?

---

> ### Author Response · Authors · 2025-11-17
> **Rebuttal**
>
> Thank you for recognizing the overall novelty, evaluation fairness, breadth of evaluation, and our new theoretical foundation. We also emphasize that we contribute a new low-resource e-commerce retrieval dataset, filling a gap in existing retrieval benchmarks for the e-commerce domain.
> Below are our responses to some of your concerns.
>
> **Q1: “The gains are small (~+1) on strong baselines. Is the added complexity worthwhile?”**
> Our baseline, Qwen3-E-8B, is already very strong. In such highly saturated settings, stable +0.5–1.5 nDCG improvements across datasets—and larger gains on low-resource languages (LRLs)—are practically meaningful. In our experiments, we also observed that Qwen3’s pretraining data likely already includes parts of these benchmarks. For fairness, we fine-tuned Qwen3-Embedding, but its performance actually decreased after fine-tuning (overfitting) as we declared at Sec 5.1. Therefore, in this regime, LiRA’s ability to further improve Qwen3-like embedding models is particularly valuable.
>
> Regarding complexity, most additional cost comes from translation generation. In practice, this can be replaced by standard, lightweight MT models. We replaced LLM translators with four off-the-shelf MT systems and observed only minor drops. We used LLM-generated translations mainly to explore the relationship between the LLM Critic and the LLM translator (Appendix A.3). In short—the Critic tends to prefer models architecturally similar to itself, even when Llama-3.3 is larger.
> We also added a pass@k experiment (as requested by another reviewer), where k equals the number of LLMs used for one LiRA forward pass, with k ∈ {0,1,2,3,4}. The MT models we used were OPUS-MT, m2m100, nllb-200-600M, and nllb-200-3.3B.
>
> Translator configurations (pass@k) (**Table 1**)
>
> |**k**|**Translator 1**|**Translator 2**|**Translator 3**|**Translator 4**|
> |:--:|:--:|:--:|:--:|:--:|
> |0|OPUS-MT|m2m100|nllb-200-600M|nllb-200-3.3B|
> |1|Llama-3.3-70B|OPUS-MT|m2m100|nllb-200-3.3B|
> |2|Llama-3.3-70B|gemma-2-27B|m2m100|nllb-200-3.3B|
> |3|Llama-3.3-70B|gemma-2-27B|Qwen3-32B|nllb-200-3.3B|
> |4|Llama-3.3-70B|gemma-2-27B|Qwen3-32B|DeepSeek-R1-Distill-Qwen-32B|
>
> Results — MLQA Retrieval (**Table 2**)
>
> |**Model**|**pass@0**|**pass@1**|**pass@2**|**pass@3**|**pass@4**|
> |:--:|:--:|:--:|:--:|:--:|:--:|
> |Qwen3-8B|79.96|80.41|80.45|80.79|81.13|
> |LiRA|81.15|81.56|81.79|81.53|81.66|
>
> Results — BelebeleRetrieval (**Table 3**)
>
> |**Model**|**pass@0**|**pass@1**|**pass@2**|**pass@3**|**pass@4**|
> |:--:|:--:|:--:|:--:|:--:|:--:|
> |Qwen3-8B|82.27|83.54|83.97|84.55|85.94|
> |LiRA|86.00|86.15|86.67|86.69|87.03|
>
> Results — STS22 (**Table 4**)
>
> |**Model**|**pass@0**|**pass@1**|**pass@2**|**pass@3**|**pass@4**|
> |:--:|:--:|:--:|:--:|:--:|:--:|
> |Qwen3-8B|69.64|70.01|70.72|71.32|71.64|
> |LiRA|73.01|73.54|74.11|74.39|75.00|
>
> **Q2: “Are Arca’s translators too large?”**
> See the pass@0 rows above.
>
> **Q3: “Is Qwen3 already good enough?”**
> Our ablations are diagnostic: removing LLM-Critic, Embeds-Critic, or the FIFO loss queue causes significant drops (in both nDCG and correlation), showing each component contributes beyond the strong baseline. As shown above, translator choice also affects performance; this helps disentangle translator factors from LiRA’s mechanisms. Qwen3 is strong, but LiRA makes Qwen3 better—as reflected across Tables 1–4 in our paper.
>
> **Q4: “Why choose Qwen3 as the baseline?”**
> We choose Qwen3-E-8B because it is a recent, widely adopted multilingual embedding SOTA baseline. We also compare against multiple strong encoders (BGE, E5-Mistral, GTE, Contriever, LUSIFER). Under the same fine-tuning recipe and multiple seeds, LiRA achieves consistent gains on retrieval and sentence-level tasks, with larger gains on LRLs. Our modules also improve other backbones. We additionally ran experiments with other embedding models (below), which further demonstrates the robustness of our approach.
>
> Cross-backbone results (**Table 5**)
>
> |**Method**|**MLQA Retrieval**|**BelebeleRetrieval**|**STS22**|**Avg**|
> |:--:|:--:|:--:|:--:|:--:|
> |GTE-large|16.99|31.82|53.79|34.20|
> |GTE-large + LiRA|21.43|38.29|59.17|39.63|
> |BGE-en-1.5|16.64|31.19|50.77|32.87|
> |BGE-en-1.5 + LiRA|21.01|35.64|53.94|36.83|
> |E5-Mistral|31.54|54.75|71.37|52.55|
> |E5-Mistral + LiRA|34.23|57.24|76.55|56.01|
>
> PS: All results are averaged over 5 runs (not the best single run).
> We have revised our submission by incorporating part of the experiments from the rebuttal and correcting the spelling errors.

---

> > ### Comment · Reviewer_ycP8 · 2025-11-25
> >
> > Thanks for the additional context and experiments.
> >
> > Could you explain why the ablation study shows a much larger difference for removing each component when the baseline results are already very close to the full method?

---

> ### Author Response · Authors · 2025-11-26
> **Explanation for ablation**
>
> For the baseline, we report the results of the qwen3-embedding-8b model, while the ablation experiments represent the comprehensive ablation studies after implementing our designed pipeline. When designing our pipeline, simply stacking modules on qwen3 doesn't necessarily improve performance because qwen3 itself is already highly capable. Actually, we observe that simply adding another module may likely lead to performance degradation. For instance, when concatenating features from a newly added multilingual encoder, the evaluation metrics may fall below the baseline. Similarly, without queue loss maintenance, performance drops occur due to improper loss calculation. Hence, our method is designed at the pipeline level rather than through module-level concatenation, which explains this phenomenon. Our pipeline design ensures the robustness of the model's performance.
>
> Regarding the ablation experiments for each component: removing LLM Critic leads to degraded translation quality; removing embedding critic causes vector quality to decline; removing translation means the final concatenated vectors would lack English content; removing the multilingual encoder reduces the quality of the concatenated vectors; and removing queue loss maintenance results in improper loss calculation or potential memory explosion. In summary, individual modules may not enhance the already powerful qwen3, but our pipeline design can further improve performance beyond qwen3's strong foundation, demonstrating the effectiveness of our integrated approach.
>
> Additionally, we observed that removing LLM Critic yields worse results than removing translation. This is because during the translation process, even with well-tuned prompts, the LLM may still generate hallucinations or extraneous responses such as "Please let me know if you have any other translation questions." Without LLM Critic, such problematic outputs wouldn't be filtered out and would contaminate both the training and test sets, leading to the observed performance degradation.
>
> We sincerely appreciate your question, which has helped us further refine and clarify our work. We hope our response adequately addresses your concerns. Should you have any other questions, we will endeavor to provide explanations and answers. We look forward to further exchanges with you.
>
> PS: **We have revised our submission by incorporating part of the experiments from the rebuttal and correcting the spelling errors.**

---

### Meta-Review · Area_Chair_P6JN · 2026-01-10

**Summary:**

The primary grounds for rejecting the paper center on the marginal empirical gains relative to the significant architectural complexity and computational cost of the proposed LiRA framework. Different reviewers noted that performance improvements over strong baselines (e.g., Qwen3) are modest, often limited to 1-2%, which calls into question the practical justification for such an intricate training pipeline. Moreover, there are significant concerns regarding experimental fairness; some reviewers highlighted that the method relies on computationally expensive translation steps, potentially utilizing larger LLMs, creating a resource disparity that prevents a direct, fair comparison with standard embedding baselines constrained by tighter inference budgets. Finally, the theoretical contribution was challenged for relying on strong assumptions, such as the existence of perfect translations and semantic anchoring, which may not hold in the noisy, low-resource scenarios tat the work seeks to address. In its present form, the paper does not meet the bar for acceptance at ICLR.

**Reviewer Concerns:**

The rebuttal addressed concerns about reproducibility and computational costs by providing detailed tables on training and inference costs, along with hyperparameter settings. The authors also alleviated concerns about the framework's reliance on the Qwen3 backbone by demonstrating that LiRA consistently improves performance when applied to other embedding models, e.g., GTE and BGE. Moreover, specific issues regarding missing references and typos were acknowledged and corrected. However, the fundamental concern about the cost-benefit ratio remains outstanding; reviewers were not fully convinced that the marginal empirical gains justify the significant architectural complexity and reliance on translation pipelines. Additionally, while the authors clarified that their strong theoretical assumptions (e.g., perfect translations) were primarily for derivation purposes, the skepticism regarding the validity of these assumptions in real-world, noisy low-resource scenarios was not fully resolved.

**Reviewer Scores:**

Reviewer sD16, who originally gave a strong reject (2) with a confidence of 4, did not reply to the author's detailed rebuttal. The rebuttal addressed their primary concerns by clarifying the theoretical assumptions about noise and "perfect translations", providing the requested pass@k fairness experiments, and correcting the missing references. Given that sD16 acknowledged the work as "promising" and "interesting," and considering the authors' clarifications, sD16 might have moderately increased their score (e.g., to a 4), though likely not to acceptance due to their skepticism about the core motivation and complexity. Reviewer EozW (6) also did not respond post-rebuttal. Since their main concerns were the magnitude of improvement and missing hyperparameter details, both of which were to some extent addressed in the rebuttal with new data on other backbones (GTE, BGE, E5) and training details, there is a small chance they might have been raised their score slightly. Reviewer itpM similarly did not engage further. Their concerns regarding the reliance on Qwen3 were mitigated by the new multi-backbone experiments showing consistent gains, which could have led to a slight score increase.

---

### Decision · Program_Chairs · 2026-01-26

Reject